# Path-specific effects for pulse-oximetry guided decisions in critical care

**Kevin Zhang** [*,†]
Columbia University
New York, USA

**Yonghan Jung** [‡]
University of Illinois Urbana-Champaign
Champaign, USA

**Divyat Mahajan**
Mila & Université de Montréal
Montreal, CA

**Karthikeyan Shanmugam**
Google DeepMind
Bengaluru, IN

**Shalmali Joshi** [†]
Columbia University
New York, USA

## Abstract

Identifying and measuring biases associated with sensitive attributes is a crucial consideration in healthcare to prevent treatment disparities. One prominent issue is inaccurate pulse oximeter readings, which tend to overestimate oxygen saturation for dark-skinned patients and misrepresent supplemental oxygen needs. Most existing research has revealed *statistical disparities* linking device measurement errors to patient outcomes in intensive care units (ICUs) without causal formalization. This study *causally* investigates how racial discrepancies in oximetry measurements affect invasive ventilation in ICU settings. We employ a causal inference-based approach using *path-specific effects* to isolate the impact of bias by race on clinical decision-making. To estimate these effects, we leverage a doubly robust estimator, propose its self-normalized variant for improved sample efficiency, and provide novel finite-sample guarantees. Our methodology is validated on semi-synthetic data and applied to two large real-world health datasets: MIMIC-IV and eICU. Contrary to prior work, our analysis reveals minimal impact of racial discrepancies on invasive ventilation rates. However, path-specific effects mediated by oxygen saturation disparity are more pronounced on ventilation duration, and the severity differs across datasets. Our work provides a novel pipeline for investigating potential disparities in clinical decision-making and, more importantly, highlights the necessity of causal methods to robustly assess fairness in healthcare.

## 1 Introduction

Bias in medical devices can perpetuate disparities by impacting healthcare decisions. For example, pulse oximeters tend to overestimate blood oxygen saturation for dark-skinned patients [45, 50, 43, 21, 42, 15]. Such discrepancies can lead to 'hidden hypoxemia,' where a patient's true arterial oxygen saturation ($SaO_2$) is dangerously low, despite a reassuringly higher peripheral $SpO_2$ reading from the oximeter. Figure 1 demonstrates this using eICU data [40], where red dots ('H.H') highlight instances where low $SaO_2$ (< 88%) is masked by higher $SpO_2$ (> 88%). Such hidden hypoxemia can result in underestimated supplemental oxygen needs and delayed clinical interventions [50, 14, 12].

Prior studies have advanced our understanding of pulse oximeter inaccuracies and associated patient outcomes. For instance, Sjoding et al. [45] and Wong et al. [50] established links between race, hidden hypoxemia, and adverse outcomes like increased organ dysfunction and mortality in large

---

[*]Currently at MIT. Contributions made when affiliated with Columbia University.

[†]Corresponding authors: {kzhang02@mit.edu, shalmali.joshi@columbia.edu}.

[‡]Contributions made when affiliated with Purdue University.

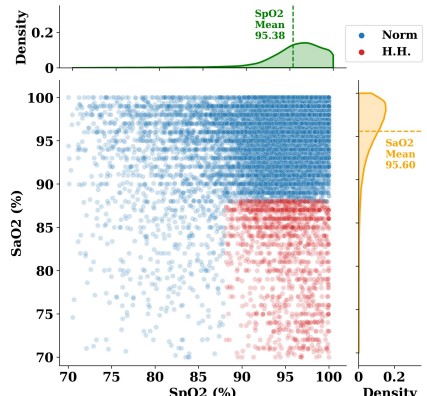

**Figure 1.** Distribution of $SpO_2$ and $SaO_2$ in eICU data. Samples with hidden hypoxemia are colored red, following Matos et al. [28].

**Table 1.** Comparison of relevant methods for path-specific causal analysis in ICU settings. ($\checkmark$: Primary focus; Part.: Some elements addressed; $\times$: Not primary focus.)

| Study | Causal Analysis | Path-Spec. Analysis | Multi-Med. | ICU Data | Finite Guar. |
|---|---|---|---|---|---|
| [45] | Assoc. | $\times$ | $\times$ | $\checkmark$ | $\times$ |
| [50] | Assoc. | $\times$ | $\times$ | $\checkmark$ | $\times$ |
| [14] | Part. | Part. | $\times$ | $\checkmark$ | $\times$ |
| [30] | $\checkmark$ | $\checkmark$ | $\checkmark$ | $\times$ | $\times$ |
| [47] | $\checkmark$ | $\checkmark$ | $\times$ | $\times$ | $\times$ |
| [49] | $\checkmark$ | $\checkmark$ | $\times$ | $\times$ | $\times$ |
| **Ours** | $\checkmark$ | $\checkmark$ | $\checkmark$ | $\checkmark$ | $\checkmark$ |

ICU datasets (e.g., MIMIC-III/IV [22]). Similarly, Gottlieb et al. [14] examined racial differences in oxygen delivery rates using regression and simple mediation analysis in MIMIC-IV, while Henry et al. [18] and Fawzy et al. [12] reported delays in hypoxemia detection and treatment eligibility. These studies rely on statistical associations or canonical mediation analysis to highlight measurement bias. However, they fail to isolate causal pathways of disparities mediated only *through* oximeter discrepancies in the presence of other mediators, which requires more advanced causal mediation analysis.

Measuring fairness using *path-specific causal analysis* is an active research area [47, 49, 30, 6, 11, 39]. These methods isolate the influence of sensitive attributes, such as race, mediated through intermediate factors. In our context (Figure 2), we aim to quantify the causal effect of race on invasive ventilation specifically *mediated by* pulse oximetry discrepancies ($V$), which is distinct from the effect via other mediators ($W$). In complex systems with *multiple mediators*, techniques for identifying and robustly estimating effects along specific pathways have been developed [49, 30].

However, Table 1 illustrates several key gaps. While sophisticated path-specific methods exist, their applications to analyze racial bias in *ICU* settings are limited, with most applications in population health or survey data [34, 13]. Furthermore, there are no *finite-sample guarantees* for path-specific effect estimators, which are crucial for generating robust evidence with potentially constrained sample sizes, like in critical care. Thus, a framework applying path-specific causal analysis for ICU problems with robust estimation strategies and theoretical backing is necessary.

To address these identified gaps, this paper makes the following primary contributions.

1. We propose a causal pipeline to detect and quantify racial disparities in the ICU. Adapting a standard fairness model (Figure 2), we use path-specific analysis to isolate the impact of race mediated through oximeter discrepancies ($V$), called the *V-specific Direct Effect (VDE)*.

2. We present robust estimators for the VDE, including a doubly robust estimator, and propose a self-normalized variant for improved sample variance, with *novel finite-sample learning guarantees*.

3. We apply our framework to analyze race-based disparities mediated by oximeter discrepancies on invasive ventilation (rate and duration) in MIMIC-IV [22] and eICU [40].

To the best of our knowledge, this is the first formal path-specific causal analysis of pulse oximeter-mediated bias on invasive ventilation using multiple mediators.

## 2 Preliminaries

We introduce preliminary notation and background in observational causal inference and fairness.

**Notation.** We use ($\mathbf{X}$, $X$, $\mathbf{x}$, $x$) to denote a random vector, variable, and their realized values, respectively. For a function $f(\mathbf{z})$, we use $\sum_{\mathbf{z}} f(\mathbf{z})$ to denote the summation/integration over discrete/continuous $\mathbf{Z}$. For a discrete vector $\mathbf{X}$, we denote $1[\mathbf{X} = \mathbf{x}]$ as an indicator function such that

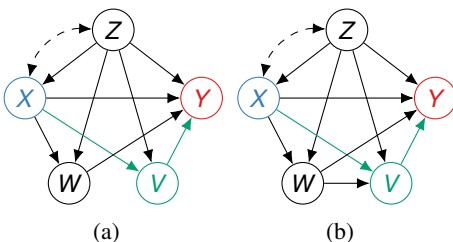

(a)        (b)

**Figure 2.** Causal diagrams for the modified standard fairness model with two mediators $W$ and $V$. For any other associations between the mediators, the VDE is not identifiable.

**Table 2.** Causal effects considered in this work. In addition to the Total Effect (TE) and Natural Direct/Indirect Effect (NDE/NIE), our application motivates the need for a path-specific effect through $V$ (VDE).

| Causal effect | Causal query |
| --- | --- |
| TE [39, 52] | $\mathbb{E}[Y_{x_1}] - \mathbb{E}[Y_{x_0}]$ |
| NDE [39, 52] | $\mathbb{E}[Y_{x_1, W_{x_0}, V_{x_0}}] - \mathbb{E}[Y_{x_0}]$ |
| NIE [39, 52] | $\mathbb{E}[Y_{x_1}] - \mathbb{E}[Y_{x_1, W_{x_0}, V_{x_0}}]$ |
| VDE (Ours) | $\mathbb{E}[Y_{x_1}] - \mathbb{E}[Y_{x_1, V_{x_0}, W_{x_1}}]$ |

$1[\mathbf{X} = \mathbf{x}] = 1$ if $\mathbf{X} = \mathbf{x}$ and 0 otherwise. $P(\mathbf{V})$ denotes a distribution over $\mathbf{V}$ and $P(\mathbf{v})$ a probability at $\mathbf{V} = \mathbf{v}$. We let $\mathbb{E}[f(\mathbf{V})]$ and $\mathbb{V}[f(\mathbf{V})]$ denote the mean and variance of $f(\mathbf{V})$ relative to $P(\mathbf{V})$, and $\|f\|_P \triangleq \sqrt{\mathbb{E}[(f(\mathbf{V}))^2]}$ as the $L_2$-norm of $f$ with $P$. We use $\widehat{f} - f = o_P(r_n)$ if $\widehat{f}$ is a consistent estimator of $f$ having rate $r_n$, and $\widehat{f} - f = O_P(r_n)$ if $\widehat{f} - f$ is bounded in probability at rate $r_n$. We will say $\hat{f}$ is $L_2$-consistent if $\|\hat{f} - f\|_P = o_P(1)$ and $\widehat{f} - f = O_P(1)$ if $\widehat{f} - f$ is bounded in probability. Let $\mathcal{D} \triangleq \{\mathbf{V}_i : i = 1, \cdots, n\}$ denote a set of $n$ samples. The empirical average of $f(\mathbf{V})$ over $\mathcal{D}$ is denoted as $\mathbb{E}_{\mathcal{D}}[f(\mathbf{V})] \triangleq (1/|\mathcal{D}|) \sum_{i:\mathbf{V}_i \in \mathcal{D}} f(\mathbf{V}_i)$.

**Structural Causal Models.** We use structural causal models (SCMs) [36] as our framework. An SCM $\mathcal{M}$ is a quadruple $\mathcal{M} = \langle \mathbf{U}, \mathbf{V}, P(\mathbf{U}), \mathcal{F} \rangle$, where $\mathbf{U}$ is a set of exogenous (latent) variables following a joint distribution $P(\mathbf{U})$, and $\mathbf{V}$ is a set of endogenous (observable) variables whose values are determined by functions $\mathcal{F} = \{f_{V_i}\}_{V_i \in \mathbf{V}}$ such that $V_i \leftarrow f_{V_i}(\mathbf{pa}_{V_i}, \mathbf{u}_{V_i})$ where $\mathbf{Pa}_{V_i} \subseteq V$ and $\mathbf{U}_{V_i} \subseteq \mathbf{U}$. Each SCM $\mathcal{M}$ induces a distribution $P(\mathbf{V})$ and a causal graph $\mathcal{G} = \mathcal{G}(\mathcal{M})$ over $\mathbf{V}$ in which directed edges exist from every variable in $\mathbf{Pa}_{V_i}$ to $V_i$ and dashed-bidirected arrows encode common latent variables. An intervention is represented using the do-operator, $\mathrm{do}(\mathbf{X} = \mathbf{x})$, which encodes the operation of replacing the original equations of $X$ (i.e., $f_X(\mathbf{pa}_X, \mathbf{u}_X)$) by the constant $x$ for all $X \in \mathbf{X}$ and induces an interventional distribution $P(\mathbf{V} \mid \mathrm{do}(\mathbf{x}))$. For any $\mathbf{Y} \subseteq \mathbf{V}$, the *potential response* $\mathbf{Y}_\mathbf{x}(\mathbf{u})$ is defined as the solution of $\mathbf{Y}$ in the submodel $\mathcal{M}_\mathbf{x}$ given $\mathbf{U} = \mathbf{u}$, which induces a *counterfactual variable* $\mathbf{Y}_\mathbf{x}$.

## 2.1 Related work

**Algorithmic Fairness.** Algorithmic fairness [16, 31, 2, 33] operationalizes computational methodologies to identify, measure, and address disparate behavior related to (automated or other) decision-making processes. One prominent line of work evaluates differential performance of machine learning models across subpopulations defined by any sensitive attribute, and proposes interventions to equalize performance or enforce specific conditional independence assumptions, such as equalized odds or opportunity [16], demographic parity [7], multicalibration [17], individual fairness [10], etc. Causal fairness is a complementary framework that assumes an underlying causal data-generating mechanism to operationalize fairness for predictive models [9, 26, 52, 39]. Principal fairness uses causal inference to enforce conditional independence of the sensitive attribute given counterfactual outcomes [20] and has been applied to coronary artery bypass grafting treatment allocation [53]. While algorithmic fairness frameworks have been used in healthcare [5, 29, 41, 38], their downstream utility has been low due to the systematic nature of statistical biases present in the data [4, 51, 46, 32].

**Path-specific Effects.** Path-specific effects are a broad class of causal effects that measure the influence of a treatment $X$ on an outcome $Y$ through specific causal pathways [37, 1, 44]. The Total Effect (TE), defined as $\mathrm{TE}(x_1, x_0) \triangleq \mathbb{E}[Y \mid \mathrm{do}(x_1)] - \mathbb{E}[Y \mid \mathrm{do}(x_0)]$, captures the influence transmitted through *all* causal paths connecting the treatment and the outcome. The total effect is decomposed into the *direct effect* and the *indirect effect*, where the latter is an effect mediated through intermediate variables. For example, in the scenario depicted by Fig. 2a, the Natural Direct Effect (NDE) [36] of $X$ on $Y$ is $\mathrm{NDE}(x_1, x_0) \triangleq \mathbb{E}[Y_{x_1, W_{x_0}, V_{x_0}}] - \mathbb{E}[Y_{x_0}]$, which captures variation in $Y$ if $X$ changes from $x_0$ to $x_1$, while the mediators $W$ and $V$ hypothetically remain the same. The Natural Indirect Effect (NIE) is $\mathrm{NIE}(x_1, x_0) \triangleq \mathbb{E}[Y_{x_1}] - \mathbb{E}[Y_{x_1, W_{x_0}, V_{x_0}}]$. This quantity represents the change in outcome due to the mediators shifting in response to the change in $X$ from $x_0$ to $x_1$.

Much of the literature has focused on estimating NDE and NIE under single-mediator settings (e.g., [47, 11]). Motivated by the presence of multiple mediators, contemporary research on path-specific analysis has broadened its scope to investigate effects transmitted along particular, often complex, paths within multi-mediator systems. For instance, effects acting through the colored path in Fig. 2a might be characterized by the *nested counterfactual* quantity $\mathbb{E}[Y_{x_1, V_{x_0}, W_{x_1}}]$ in statistics and algorithmic fairness [49, 30, 52, 6, 8]. Specifically, [30] developed a robust estimator for measuring path-specific effects transmitted through a particular mediator in the presence of other mediators.

# 3 Pulse-oximetry bias using path-specific analysis

We define path-specific effects within the standard fairness model (SFM), illustrated by the DAG in Figure 2, which represents a broad class of clinical problems. In our setup, $X$ is a binary indicator of race, $Y$ represents ventilation-related outcomes, and $Z$ denotes the pre-admission statistics. The mediator $W$ represents post-admission measurements, while the mediator $V$ includes variables related to oxygen saturation, such as the peripheral oxygen saturation reading from a pulse oximeter $SpO_2$, the more accurate arterial blood gas measurement ($SaO_2$), and the discrepancy $\Delta = SpO_2 - SaO_2$. We examine two primary outcomes: (1) the rate of invasive ventilation, defined as the proportion of patients who receive any invasive ventilation procedure during their ICU stay, and (2) the duration of invasive ventilation, measured as the length of the first invasive ventilation event.

We study the direct effect of $X$ on $V$ which then affects $Y$, termed the *V-specific Direct Effect* (VDE):

$$\text{VDE}(x_1, x_0) \triangleq \mathbb{E}[Y_{x_1}] - \mathbb{E}[Y_{x_1, V_{x_0}, W_{x_1}}] \tag{1}$$

This path-specific effect, which isolates the influence mediated through $V$ given $W_{x_1}$ (the mediators $W$ as they would be under exposure $x_1$), corresponds to estimands discussed in prior work (e.g., Miles et al. [30]). Within our SFM, we specifically term this as the VDE. As an example, suppose $x_0 = \texttt{White}$, $x_1 = \texttt{Black}$, and $Y$ is the rate of invasive ventilation. The VDE captures the difference in expected invasive ventilation rate ($Y$) if post-admission measurements ($W$) *and* the ventilation decision were made as if the patient were Black ($x_1$), while oxygen saturation readings ($SpO_2$, $SaO_2$, $\Delta$) were measured as if the patient were White ($x_0$). This effect isolates treatment rate heterogeneity due to discrepancies in oxygen saturation measurements. The SFM in Figure 2 captures plausible associations between the mediators $W$ and $V$ in which the VDE is identifiable (see proof in Appendix A). Table 2 contextualizes and compares all causal effects of interest within our framework.

## 3.1 Estimating path-specific effects

This section presents identifiability conditions for the VDE (Eq. (1)) and then develops its doubly robust estimator. Our approach integrates key insights from Miles et al. [30], who addressed similar path-specific effects often under parametric assumptions, and Jung et al. [25], who provided a general non-parametric framework for complex estimands involving conditional expectations. By synthesizing these, we develop a doubly robust VDE estimator that, crucially, does not require parametric modeling. This allows for the flexible use of machine learning methods to estimate these functions, which improves robustness to model misspecification. Under the SFM in Figs. (2a, 2b), the VDE is identifiable as follows.

**Proposition 1 (Identifiability [30]).** *Under the SFMs in Fig. 2, the VDE $\mathbb{E}[Y_{x_1}] - \mathbb{E}[Y_{x_1, V_{x_0}, W_{x_1}}]$ is identifiable and is given as follows: $\mathbb{E}[Y_{x_1}] = \sum_z \mathbb{E}[Y \mid x_1, z]P(z)$ and*

$$\mathbb{E}[Y_{x_1, V_{x_0}, W_{x_1}}] = \sum_{w,v,z} \mathbb{E}[Y \mid x_1, w, v, z]P(v \mid x_0, w, z)P(w \mid x_1, z)P(z). \tag{2}$$

We focus on estimating the nested counterfactual term of the VDE, $\mathbb{E}[Y_{x_1, V_{x_0}, W_{x_1}}]$ represented in Eq. (2), since estimating $\mathbb{E}[Y_{x_1}]$ using the back-door adjustment [35] is well-known.

To construct the estimator, we will parameterize Eq. (2) using the following:

**Definition 1 (Nuisance Parameters [30]).** *Two sets of nuisance parameters $\boldsymbol{\mu}_0 \triangleq \{\mu_0^1, \mu_0^2, \mu_0^3\}$ and $\boldsymbol{\pi}_0 \triangleq \{\pi_0^1, \pi_0^2, \pi_0^3\}$ are defined as shown in the following table, where $\check{\mu}_0^3 \triangleq \mu_0^3(V, W, x_1, Z)$ and $\check{\mu}_0^2 \triangleq \mu_0^2(W, x_0, Z)$.*

| Regression Parameters $\mu_0$ | | Importance Sampling Parameters $\pi_0$ | |
|---|---|---|---|
| $\mu_0^3(V, W, X, Z)$ | $\triangleq \mathbb{E}[Y \mid V, W, X, Z]$ | $\pi_0^3(V, W, X, Z)$ | $\triangleq \frac{P(V\mid x_0, W, Z)}{P(V\mid X, W, Z)} \frac{1[X=x_1]}{P(X\mid Z)}$ |
| $\mu_0^2(W, X, Z)$ | $\triangleq \mathbb{E}[\check{\mu}_0^3 \mid W, X, Z]$ | $\pi_0^2(W, X, Z)$ | $\triangleq \frac{P(W\mid x_1, Z)}{P(W\mid X, Z)} \frac{1[X=x_0]}{P(X\mid Z)}$ |
| $\mu_0^1(X, Z)$ | $\triangleq \mathbb{E}[\check{\mu}_0^2 \mid X, Z]$ | $\pi_0^1(X, Z)$ | $\triangleq \frac{1[X=x_1]}{P(X\mid Z)}$ |

Then, the VDE in Eq. (2) can be parametrized as follows.

**Lemma 1 (Parametrization).**

$$Eq.\ (2) = \mathbb{E}[Y \times \pi_0^3(V, W, X, Z)] = \mathbb{E}[\mu_0^1(x_1, Z)] = \mathbb{E}[\mu_0^i \times \pi_0^i], \ for\ i = 1, 2, 3. \tag{3}$$

Among all parametrizations of Eq. (2), we use the one with the double robustness property:

$$\varphi((Y, V, W, X, Z); \boldsymbol{\mu}_0, \boldsymbol{\pi}_0) \triangleq \pi_0^3(V, W, X, Z)\{Y - \mu_0^3(V, W, X, Z)\} \tag{4}$$

$$+ \pi_0^2(W, X, Z)\{\mu_0^3(V, W, x_1, Z) - \mu_0^2(W, X, Z)\} \tag{5}$$

$$+ \pi_0^1(X, Z)\{\mu_0^2(W, x_0, Z) - \mu_0^1(X, Z)\} + \mu_0^1(x_1, Z). \tag{6}$$

The functional $\varphi$ is a valid representation since $\mathbb{E}[\varphi((Y, V, W, X, Z); \boldsymbol{\mu}_0, \boldsymbol{\pi}_0)] =$ Eq. (2). Moreover, $\varphi$ exhibits the following robustness property:

**Lemma 2 (Double Robustness Property).** *Let* $\mathbf{V} = (Y, V, W, X, Z)$. *For any arbitrary vectors* $\boldsymbol{\mu}, \boldsymbol{\pi}$*, the functional* $\varphi$ *satisfies the following double robustness property:*

$$\mathbb{E}[\varphi(\mathbf{V}; \boldsymbol{\mu}_0, \boldsymbol{\pi}_0)] - \mathbb{E}[\varphi(\mathbf{V}; \boldsymbol{\mu}, \boldsymbol{\pi})] = \sum_{i=1}^{3} O_P(\|\mu^i - \mu_0^i\|_P \|\pi^i - \pi_0^i\|_P). \tag{7}$$

Eq. (7) shows debiasedness because whenever $\hat{\mu}^i, \hat{\pi}^i$ converges to $\mu_0^i, \pi_0^i$ with rate $n^{-1/4}$, then the error in $\varphi$ converges at a much faster $n^{-1/2}$ rate. We construct the following doubly robust estimator.

**Definition 2 (Doubly Robust VDE Estimator).** *Given a sample* $\mathcal{D} \overset{i.i.d.}{\sim} P$*, the doubly robust VDE estimator* $\hat{\psi}$ *is constructed as follows:*

1. *(Sample-Splitting) Take any $L$-fold random partition for the dataset* $\mathcal{D} \triangleq (\mathbf{V}_1, \cdots, \mathbf{V}_n)$*; i.e.,* $\mathcal{D} = \cup_{\ell=1}^{L} \mathcal{D}_\ell$ *where the size of the partitioned dataset* $\mathcal{D}_\ell$ *is equal to* $n/L$.

2. *(Learning by Partitions) For each* $\ell = 1, 2, \cdots, L$*, construct the estimator* $\hat{\mu}_\ell^i, \hat{\pi}_\ell^i$ *using* $\mathcal{D} \setminus \mathcal{D}_\ell$ *for* $i = 1, 2, 3$ *and compute* $\hat{\psi}_\ell \triangleq \mathbb{E}_{\mathcal{D}_\ell}[\varphi(Y, V, W, X, Z); \hat{\boldsymbol{\mu}}, \hat{\boldsymbol{\pi}}]$

3. *(Aggregation) The one-step estimator* $\hat{\psi}$ *is an average of* $\{\hat{\psi}_\ell\}_{\ell=1}^{L}$*; i.e.,* $\hat{\psi} \triangleq \frac{1}{L}(\hat{\psi}_1 + \cdots, \hat{\psi}_L)$.

Following the analysis from [Theorem 4 in 25], we provide novel finite-sample learning guarantees:

**Theorem 2 (Finite-Sample Analysis).** *Suppose* $\hat{\mu}_\ell^i, \mu_0^i < \infty$ *and* $0 < \hat{\pi}_\ell^i, \pi_0^i < \infty$ *almost surely for* $i = 1, 2, 3$*. Suppose the third moment of* $\varphi$ *exists. Let* $R_1 \triangleq (1/L) \sum_{\ell=1}^{L} (\mathbb{E}_{\mathcal{D}_\ell}[\hat{\varphi}_\ell] - \mathbb{E}_P[\varphi_0])$. *Let* $\rho_0^2 \triangleq \mathbb{V}[\varphi_0]$. *Let* $\kappa_0^3 \triangleq \mathbb{E}[|\varphi|^3]$. *Let* $\Phi(x)$ *denote the standard normal CDF. Then,*

$$\hat{\psi} - \psi_0 = R_1 + \frac{1}{L} \sum_{\ell=1}^{L} \sum_{i=1}^{3} \mathbb{E}[\{\hat{\mu}_\ell^i - \mu_0^i\}\{\pi_0^i - \hat{\pi}_\ell^i\}], \tag{8}$$

*where, with probability greater than* $1 - \epsilon$,

$$R_1 \leq \sqrt{\frac{2}{\epsilon}} \left( \sqrt{\frac{L\rho_0^2}{n}} + \sqrt{\sum_{\ell=1}^{L} \frac{L\|\hat{\varphi}_\ell - \varphi_0\|_P^2}{n}} \right), \quad and \tag{9}$$

$$\left| Pr\left( \frac{\sqrt{L}}{\sqrt{n}\rho_0} R_1 \right) - \Phi(x) \right| \leq \frac{1}{\sqrt{2\pi}} \sqrt{\frac{1}{\epsilon} \sum_{\ell=1}^{L} \frac{L\|\hat{\varphi}_\ell - \varphi_0\|_P^2}{n}} + \frac{0.4748\kappa_0^3}{\rho_0^3 \sqrt{n}}. \tag{10}$$

Theorem 2 demonstrates that the error can be decomposed into two terms. The term $R_1$ closely approximates a standard normal distribution variable, and the remaining term exhibits the doubly-robustness behavior. This is formalized in the corresponding asymptotic analysis.

**Corollary 2 (Asymptotic error).** *Assume $\mu_0^i, \hat{\mu}_\ell^i < \infty$ and $0 < \pi_0^i, \hat{\pi}_\ell^i < \infty$ almost surely. Suppose $\hat{\mu}_\ell^i$ and $\hat{\pi}_\ell^i$ are $L_2$-consistent. Then,*

$$\hat{\psi} - \psi_0 = R_1 + \frac{1}{L} \sum_{\ell=1}^{L} \sum_{i=1}^{3} O_P(\|\hat{\mu}_\ell^i - \mu_0^i\| \, \|\pi_0^i - \hat{\pi}_\ell^i\|),$$

*and $\sqrt{n} R_1$ converges in distribution to Normal$(0, \rho_{k,0}^2)$.*

To estimate $\hat{\pi}^i \in \hat{\boldsymbol{\pi}}$, we can rewrite the nuisance expressions using Bayes' rule to avoid computing densities in high dimensions. However, estimation can still be challenging because propensities in the denominator may cause the variance to explode. Since $\mathbb{E}[\pi_0^i] = 1$, we can improve the estimation stability using self-normalization (SN) by setting $\hat{\pi}_{\text{SN}}^i \leftarrow \hat{\pi}^i / \mathbb{E}_{\mathcal{D}}[\hat{\pi}^i]$. This technique is known to reduce the variance [19]. The self-normalized estimator for the NDE/NIE is outlined in Appendix B. Theoretical results follow analogously for the self-normalized doubly-robust estimator.

# 4 Experiments

We evaluate our proposed canonical and self-normalized doubly robust estimators for computing the fairness effects in Table 2 across synthetic, semi-synthetic, and real-world settings. Like most causal inference literature, we validate our methodology using synthetic and semi-synthetic data, which provide access to the true data-generating mechanisms and counterfactual outcomes.

We then apply our methodology to two real-world ICU datasets to estimate the path-specific effect of race (specifically mediated by discrepancies in pulse oximetry measurements) on the rate and duration of invasive treatment. Note that unlike previous studies which examine patient outcomes, we study decision-making regarding invasive ventilation procedures in the ICU.

For all experiments, we perform sample-splitting with $L = 5$ folds and clip propensities within the range $[\varepsilon, 1 - \varepsilon]$, where $\varepsilon = 10^{-4}$. All propensity and outcome models are trained using XGBoost. Details on hyperparameter selection are provided in Appendix D and code credits are in Appendix L.

## 4.1 Synthetic data

In this experiment, we demonstrate the finite-sample behavior of the canonical and self-normalized doubly robust estimator. We validate convergence of causal estimands using a linear synthetic setting, where $Z$ and $W$ are multi-dimensional continuous vectors.

We generate samples from a linear SCM with randomly initialized weights, where $Z, V$ are 3-dimensional, $W$ is 10-dimensional, and $X, Y$ are single-dimensional. These dimensionalities are chosen to be similar to the real-world data. We estimate all causal queries for computing fairness effects in Table 2, namely: $\mathbb{E}[Y_{x_0}]$, $\mathbb{E}[Y_{x_1}]$, $\mathbb{E}[Y_{x_1, M_{x_0}}]$, and $\mathbb{E}[Y_{x_1, V_{x_0}, W_{x_1}}]$. To show the convergence, we vary the sample size from 1,000 to 32,000, which roughly matches the number of eICU samples.

In Figures 3a and 3b, we report the mean and 95% confidence interval of the relative error for each causal query across 100 bootstraps using the standard doubly robust (non-SN) and self-normalized (SN) estimator. While both estimators converge to the true causal quantities, the errors without self-normalization exhibit significantly higher variance. In Appendix E, we demonstrate that this difference primarily occurs because the empirical mean of the nuisance parameters deviates from one.

## 4.2 ICU cohort selection

To conduct semi-synthetic/real-world experiments, we use two large, publicly available critical care datasets: the eICU Collaborative Research Database (eICU) with ICU admissions from hospitals across the continental U.S. [40], and Medical Information Mart for Intensive Care (MIMIC-IV), with ICU data from the Beth Israel Deaconess Medical Center in Boston [22].

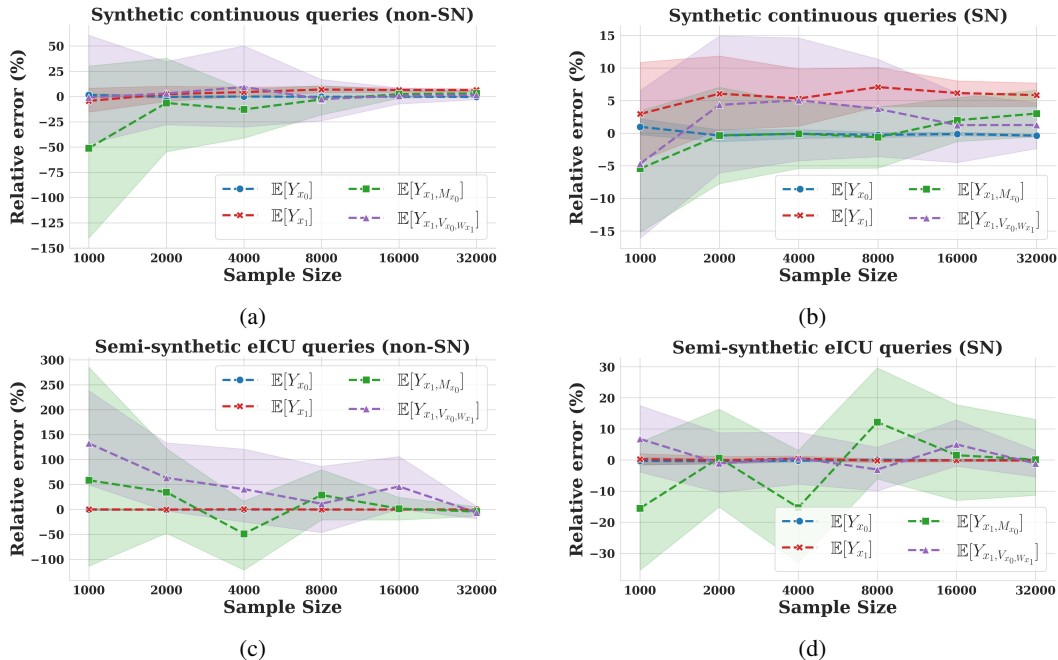

**Figure 3.** Convergence of the relative error of causal queries using the canonical (left column) and self-normalized (right column) doubly robust estimator on a continuous synthetic setting (top row) and semi-synthetic eICU data (bottom row). The plots show the mean and 95% confidence interval over 100 bootstrap iterations. The estimates using self-normalization exhibit significantly lower variance.

We are interested in investigating the causal effect of race on invasive ventilation rate and duration mediated by oxygen saturation discrepancy. If a patient undergoes multiple intubations over the course of a single ICU admission, only the first is considered. We treat gender, age, and comorbidity scores as confounders, representing patient characteristics available before ICU admission and prior to any clinical procedures or measurements. Comorbidity is quantified using the Charlson Comorbidity Index and the pre-ICU OASIS score, both of which summarize baseline health status.

The mediators $W$ consist of post-admission measurements: arterial blood gas (ABG) results excluding $SpO_2$ and $SaO_2$, laboratory values such as creatinine and lactate, and periodic vital signs (e.g., temperature, respiratory rate). The $V$ mediators include any $SpO_2$ and $SaO_2$ values within the range $[70\%, 100\%]$, as well as the measurement discrepancy $\Delta = SpO_2 - SaO_2$. For computing $\Delta$, we match each $SpO_2$ reading with an $SaO_2$ measurement taken within the next five minutes if available.

Note that our $V$ mediators include not only the discrepancy, but also $SaO_2$ and $SpO_2$, because race influences the discrepancy through oxygen saturation measurements. Although $V$ includes patient variables that are not intrinsically unfair, incorporating all oximetry-related variables captures the mechanism by which racial bias in pulse oximetry mediates clinical decisions more comprehensively.

To summarize measurement trajectories over time, we compute an exponentially weighted average from ICU admission up to the time of first ventilation (or discharge if the patient was not intubated). For a sequence of observations $x_1, \ldots, x_n$ taken $t_1, \ldots, t_n$ minutes before the first ventilation or discharge, the aggregated value is $\overline{x} = \sum_i w_i x_i / \sum_i w_i$, where $w_i = \exp(-\gamma t_i)$ is an exponentially decaying weight and $\gamma$ is a smoothing parameter. We restrict the analysis to stays lasting at least 24 hours and with at least one recorded measurement of $SpO_2$, $SaO_2$, and a matched discrepancy value. The final cohort consists of 37,222 admissions from eICU and 4,897 admissions from MIMIC-IV. Figure 4 presents the distribution of ICU stays by race, ventilation status, and treatment duration. Details on the distribution of pre-admission severity scores are included in Appendix C.

### 4.3 Semi-synthetic eICU experiments

We create a semi-synthetic cohort by leveraging real-data patterns to construct a synthetic SCM. Semi-synthetic data allows us to evaluate our methodology in a more realistic setting while retaining access to ground-truth effects for error analysis.

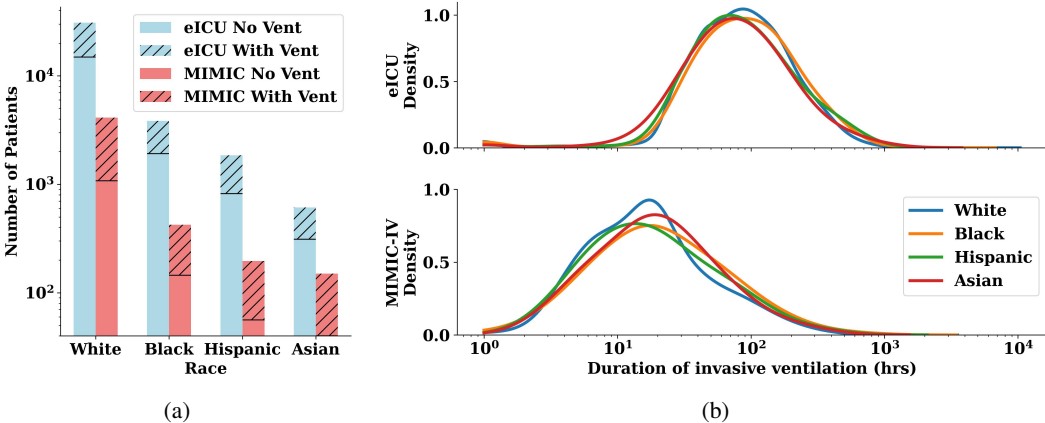

**Figure 4.** Distribution of samples in the eICU and MIMIC-IV datasets, showing (a) number of patients by race and ventilation status and (b) invasive ventilation duration, stratified by race. The baseline duration of ventilation is higher for eICU patients, and is associated with greater pre-admission severity (see Appendix C).

To generate the cohort, we use the real-world eICU dataset to train separate XGBoost models to predict each variable in the set $\{X, W, V, Y\}$ given its observed parents. In the semi-synthetic setting, $Y$ is a binary variable indicating whether the patient received an invasive ventilation procedure during the stay. These trained models serve as the true causal mechanisms when generating samples. We use the eICU dataset due to its larger sample size compared to MIMIC-IV.

Following the setup in the synthetic experiment, we compute the relative error for $\mathbb{E}[Y_{x_0}]$, $\mathbb{E}[Y_{x_1}]$, $\mathbb{E}[Y_{x_1, M_{x_0}}]$, and $\mathbb{E}[Y_{x_1, V_{x_0}, W_{x_1}}]$ across varying sample sizes in Figures 3c and 3d. Similar to the synthetic experiments, our proposed self-normalized variant exhibits significantly lower variance in a healthcare-grounded setting with complex relationships between variables. This behavior is consistent with results using MIMIC-based semi-synthetic data, which we include in Appendix G. Additionally, we compare our estimators with state-of-the-art effect estimation baselines in Appendix I and demonstrate their robustness to imbalance between $x_0$ and $x_1$ subpopulations in Appendix J.

Results on semi-synthetic data support our approach over other methods in Table 1. Most prior work only allow for a single mediator, which could be applicable in other ICU settings, but is limited for our pulse oximetry application because relevant patient variables beyond oxygen discrepancy must be taken into consideration for valid and unbiased estimates of fairness. While Miles et al. [30] use multiple mediators, their estimator is conceptually analogous to our non-self-normalized variant, which exhibits high variance in small-sample settings like ours. Based on these considerations, we adopt the self-normalized estimator for quantifying path-specific effects in real-world data.

### 4.4 Real-world eICU and MIMIC-IV experiments

We estimate the effects in Table 2 using $x_0 = \text{White}$ as the baseline (reflecting the majority demographic of the cohort) and $x_1 = \text{Black}$. Additionally, we compute the NIE as if we entirely ignored $W$ and denote this effect as $\text{NIE}^*$. This alternative effect allows for comparison with existing research, as prior studies have quantified the effect of race mediated by measurement discrepancy without accounting for other post-admission mediators (denoted as $W$ in our setup). For example, Gottlieb et al. [14] used $\text{NIE}^*$ in the context of pulse oximetry to quantify the effect of the discrepancy on oxygen supplementation levels, but did not account for the impact of other potential mediators on the outcome. In contrast, our methodology explicitly models these mediators using $W$.

We compute all causal effects over 500 bootstrap iterations and report the mean and 95% confidence intervals. Figure 5 shows the effects for the rate and duration of ventilation. A positive-valued effect indicates that the corresponding pathway contributes to more frequent or longer invasive ventilation treatments for Black patients relative to White patients. We include the results using the canonical doubly robust estimator in Appendix H. All of our analysis and conclusions are consistent between the self-normalized and the canonical doubly robust estimators, with some slight differences in the effect magnitudes.

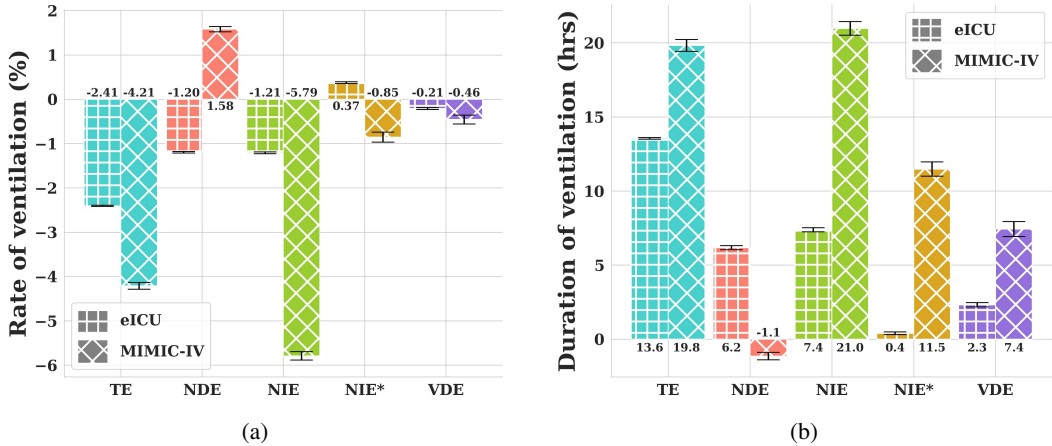

**Figure 5.** Average causal fairness measures across 500 bootstraps using the self-normalized estimator for the (a) rate and (b) duration of invasive ventilation on eICU and MIMIC-IV data. Colors represent different effects and the numerical label for each bar indicates the mean across all bootstraps. Error bars show 95% confidence intervals. Positive values indicate a higher rate or longer duration of invasive ventilation for Black patients relative to White patients.

**Analysis of ventilation rates.** Figure 5a suggests that the overall total effect varies across the two datasets, which indicates baseline practice differences. While the negative TE indicates baseline ventilation rates are higher for White patients compared to Black patients, these differences do not indicate unfairness without studying the decomposed effects. A negative NDE in eICU versus a positive NDE in MIMIC-IV suggests direct discrepancy adjusted only for baseline patient confounders, but not ICU-specific patient condition. Much of the total variation in both eICU and MIMIC-IV is dominated by the NIE, suggesting potential differences in White vs. Black patient health conditions influencing ventilation rates. The VDE for invasive ventilation rates for both eICU ($-0.21$ percentage points, 95% CI [$-0.23$ to $-0.19$]) and MIMIC-IV ($-0.46$ percentage points, 95% CI [$-0.56$ to $-0.36$]) datasets is relatively small, indicating low unfairness mediated by oxygen saturation discrepancy. In addition, the NIE* shows that ignoring $W$ mediators for this problem can exacerbate the estimated scale of unfairness and potentially flip the direction of disparity.

**Analysis of ventilation duration.** Figure 5b shows significant differences in the duration of ventilation between eICU and MIMIC-IV, which is largely indicative of baseline distribution shifts in both ICU datasets (see Appendix C). The flipped signs of NDE specify that across multicenter eICU data, Black patients are ventilated for longer, while in MIMIC-IV, White patients are ventilated for longer, when adjusting for pre-admission covariates. The NIE is the primary pathway influencing the total effect, however the effect severity drastically shifts when considering only a subset of mediators (i.e. NIE*), which suggests the need to decompose NIE further to assess VDE. We observe slightly longer durations of invasive ventilation among Black over White patients in eICU (2.3 hrs, 95% CI [2.2 to 2.5]) and significantly longer durations in MIMIC-IV (7.4 hrs, 95% CI [6.9 to 7.9]), mediated by pulse oximetry discrepancies. The MIMIC-IV bias in duration is broadly in line with some reported differences in levels of end-of-life care by race [24, 23], but requires further investigation to attribute observed VDE differences to meaningful causes.

To better understand how oxygen discrepancy influences the VDE, we present the effect conditioned on both the discrepancy $\Delta$ and the $SpO_2$ value. Training a regression model for the VDE proved challenging due to small sample sizes in certain regions; therefore, we compute averages over discrete binned intervals of $\Delta$ and $SpO_2$. The conditional effects are shown in Figure 6.

While MIMIC-IV samples are sparse, the eICU conditional VDE suggests that under hypoxemia ($SpO_2 \leq 85\%$ and $\Delta \geq 0\%$), higher rates of ventilation tend to occur for Black compared to White patients. However, the effect flips when $85\% \leq SpO_2 \leq 90\%$ and $10\% \leq \Delta \leq 20\%$. The conditional VDE for ventilation duration is relatively small and uniform when there is little oxygen discrepancy ($|\Delta| \leq 10\%$). Meanwhile, Black patients experience significantly longer ventilation times when $90\% \leq SpO_2 \leq 95\%$ and $10\% \leq \Delta \leq 20\%$ (hidden hypoxemia) in both datasets. This pattern also persists when $80\% \leq SpO_2 \leq 90\%$ and $-20\% \leq \Delta \leq -10\%$ (moderate hypoxemia), but the effect flips in the region $90\% \leq SpO_2 \leq 95\%$ for eICU samples.

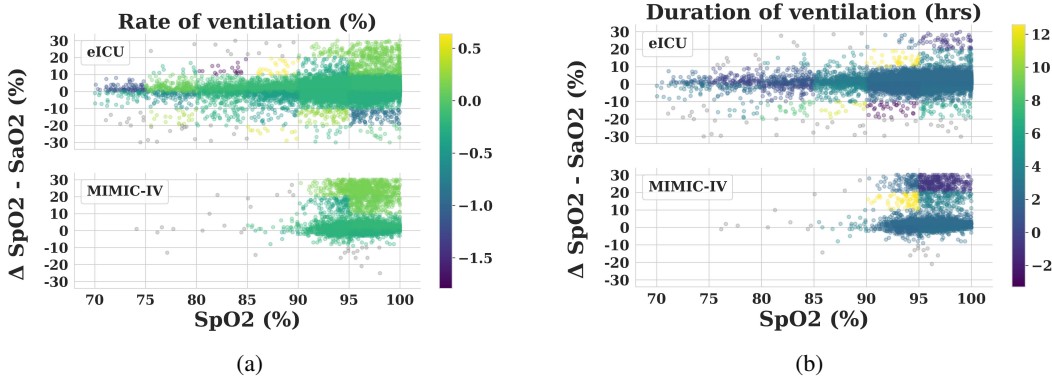

**Figure 6.** VDE conditioned on the discrepancy $\Delta$ and SpO$_2$ for the (a) rate and (b) duration of invasive ventilation on eICU and MIMIC-IV data. Due to small sample sizes in certain regions, we take the average VDE over binned intervals of $(\Delta, \text{SpO}_2)$. Regions with fewer than 20 units are shown in gray.

## 5 Discussion

In this work, we introduce the $V$-specific Direct Effect (VDE) to quantify racial biases in critical care decision-making that are mediated by measurement discrepancies from pulse oximeter devices. While previous studies have documented disparate outcomes for Black patients, our approach is the first to apply a path-specific causal framework to examine heterogeneity by race in a clinically actionable healthcare process. We develop a doubly robust estimator and a self-normalized variant for the VDE, provide favorable finite-sample guarantees, and demonstrate strong empirical performance on both synthetic and semi-synthetic health data.

In two publicly available ICU datasets, we find negligible disparities in invasive ventilation rates indicating slightly less frequent treatment for Black patients relative to White patients mediated by oxygen saturation discrepancy. Both datasets also show longer treatment durations for Black compared to White patients with different effect severities. We consistently observe that the canonical NIE exacerbates the magnitude of disparity while the NIE* may potentially flip the direction. This highlights the necessity of our path-specific causal framework to accurately examine fairness.

For assessing clinical decision-making, our findings indicate that bias arising from pulse oximetry measurements primarily affects the duration of invasive ventilation rather than initiation of treatment. This pattern suggests that clinicians may effectively integrate additional patient information beyond oxygen saturation when deciding whether to ventilate. Alternatively, the bias may manifest through pathways not captured within our current framework, such as through unmeasured variables or outcomes. Understanding the mechanisms underlying such bias remains an active area of research.

**Limitations.** As with most causal inference methods, our framework relies on a specified causal graph and thus depends on assumptions about the data-generating process that can be challenging to verify in practice. Consequently, it shares common limitations related to potential misspecification of the causal structure. For example, if there is strong reason to believe that $W$ and $V$ have a latent common cause in a given application, any estimated path-specific effect for the VDE would be inaccurate because the true effect is not identifiable.

Our causal graph reflects generally plausible mechanisms by ensuring clinically relevant quantities reported in the datasets are modeled, while making the minimal assumptions required for identifiability of the causal effects of interest. The graph explicitly captures these underlying assumptions, which is an important advantage highlighted in critical care literature [27].

We only consider the first invasive ventilation and do not explicitly model temporality in our analysis. Additionally, sparse SaO$_2$ measurements limit the frequency of matched oxygen saturation discrepancy measurements, further affecting sample size. However, we emphasize that this is a limitation with the dataset and not inherent to our method.

**Broader Impact.** Several clinical problems can be framed using the multiple-mediation analyses. Our contributions are intended to provide a practical algorithm for computing particular path-specific effects more broadly. The methods provide a more nuanced understanding of heterogeneity in current healthcare practices.

## Acknowledgements

KZ and SJ acknowledge partial support from Google Inc. DM acknowledges support via FRQNT doctoral scholarship (`https://doi.org/10.69777/354785`) for his graduate studies. The authors thank Jack Gallifant and Leo Celi for preliminary discussions, which helped inform the problem setup. Any opinions, findings, conclusions, or recommendations in this manuscript are those of the authors and do not reflect the views, policies, endorsements, expressed or implied, of any aforementioned funding agencies/institutions.

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

# A Proofs

## A.1 Identifiability of VDE in Figures (2a, 2b)

Figures (2a, 2b) imply the following conditional independences between counterfactual variables:

$$V_{x_0,w} \perp\!\!\!\perp W_{x_1} \mid Z, \tag{11}$$
$$W_x \perp\!\!\!\perp X \mid Z \tag{12}$$
$$V_{x,w} \perp\!\!\!\perp W, X \mid Z \tag{13}$$
$$Y_{x_1,v} \perp\!\!\!\perp V_{x_0,w} \mid Z, W_{x_1}. \tag{14}$$

These conditional independences lead to the following derivation:

$$\mathbb{E}[Y_{x_1,V_{x_0,W_{x_1}}}]$$
$$\sum_z \mathbb{E}[Y_{x_1,V_{x_0,W_{x_1}}} \mid z]P(z)$$
$$= \sum_{v,w,z} \mathbb{E}[Y_{x_1,v} \mid V_{x_0,w}=v, W_{x_1}=w, z]P(V_{x_0,w}=v, W_{x_1}=w \mid z)P(z)$$
$$= \sum_{v,w,z} \mathbb{E}[Y_{x_1,v} \mid V_{x_0,w}=v, W_{x_1}=w, z]P(V_{x_0,w}=v \mid W_{x_1}=w, z)P(W_{x_1}=w \mid z)P(z)$$
$$= \sum_{v,w,z} \mathbb{E}[Y_{x_1,v} \mid V_{x_0,w}=v, W_{x_1}=w, z]P(v \mid x_0, w, z)P(w \mid x_1, z)P(z)$$
$$= \sum_{v,w,z} \mathbb{E}[Y \mid x_1, v, w, z]P(v \mid x_0, w, z)P(w \mid x_1, z)P(z),$$

where

1. Since $W_x \perp\!\!\!\perp X \mid Z$, we have $P(W_{x_1}=w \mid z) = P(w \mid x_1, z)$.

2. Since $V_{x_0,w} \perp\!\!\!\perp W_{x_1} \mid Z$, we have $P(V_{x_0,w}=v \mid W_{x_1}=w, z) = P(V_{x_0,w}=v \mid z)$.

3. Since $V_{x_0,w} \perp\!\!\!\perp W, X \mid Z$, we have $P(V_{x_0,w}=v \mid z) = P(v \mid x_0, w, z)$.

4. Since $Y_{x_1,v} \perp\!\!\!\perp V_{x_0,w} \mid Z, W_{x_1}$, we have

$$\mathbb{E}[Y_{x_1,v} \mid V_{x_0,w}=v, W_{x_1}=w, z] = \mathbb{E}[Y_{x_1,v} \mid W_{x_1}=w, z]$$
$$= \mathbb{E}[Y \mid \text{do}(x_1,v), w, z] = \mathbb{E}[Y \mid x_1, v, w, z].$$

## A.2 Non-identifiability of VDE in Figures (7a, 7b)

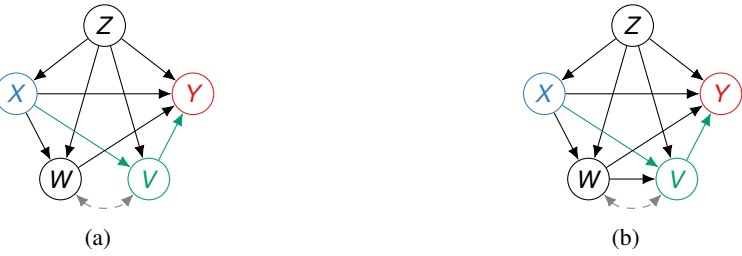

(a)                                    (b)

**Figure 7.** Causal diagrams for the modified standard fairness model with two mediators $W$ and $V$ with latent common causes that render the green path-specific effect unidentifiable.

Whenever there exist unmeasured confounders between $W$ and $V$, the VDE is not identifiable. Specifically, we can write the counterfactual nested term $P(Y_{x_1,V_{x_0,W_{x_1}}} = y)$ as follows using the

identification algorithm proposed by [8]:

$$P(Y_{x_1, V_{x_0, W_{x_1}}} = y) = \sum_v P(Y_{x_1, v} = y, V_{x_0, W_{x_1}} = v) \tag{15}$$

$$= \sum_{v,w} P(Y_{x_1, v, w} = y, V_{x_0, w} = v, W_{x_1} = w) \tag{16}$$

$$= \sum_{v,w,x,z} P(Y_{x_1, v, z, w} = y, V_{x_0, w, z} = v, W_{x_1, z} = w, X_z = x, Z = z) \tag{17}$$

$$= \sum_{v,w,x,z} P(Y_{x_1, v, z, w} = y) P(V_{x_0, w, z} = v, W_{x_1, z} = w) P(X_z = x) P(Z = z). \tag{18}$$

This reduces the problem to identifying $P(V_{x_0, w, z} = v, W_{x_1, z} = w)$. By [Theorem 3 in 8], this term is not identifiable because of inconsistency (where $X = 0$ in the counterfactual $V_{x_0, w, z}$ while $X = 1$ in $W_{x_1, z}$).

### A.3 Proof of Lemma 1

We note that

$$\mu_0^2(W, X, Z) \triangleq \mathbb{E}[\mu_0^3(V, W, x_1, Z) \mid W, X, Z] \tag{19}$$

$$= \sum_v \mu^3(v, W, x_1, Z) P(v \mid W, X, Z) \tag{20}$$

$$= \sum_v \mathbb{E}[Y \mid v, W, x_1, Z] P(v \mid W, X, Z) \tag{21}$$

and

$$\mu_0^1(X, Z) \triangleq \mathbb{E}[\mu_0^2(W, x_0, Z) \mid X, Z] \tag{22}$$

$$= \sum_w \mu_0^2(w, x_0, Z) P(w \mid X, Z) \tag{23}$$

$$= \sum_w \sum_v \mathbb{E}[Y \mid v, w, x_1, Z] P(v \mid w, x_0, Z) P(w \mid X, Z). \tag{24}$$

Therefore,

$$\mathbb{E}[\mu_0^1(x_1, Z)] = \text{Eq. (2)}. \tag{25}$$

Furthermore,

$$\mathbb{E}[\pi_0^3(V, W, X, Z) Y] = \mathbb{E}[\pi_0^3(V, W, X, Z) \mu_0^3(V, W, X, Z)] \tag{26}$$

$$= \sum_{v,w,z} \mu^3(v, w, x_1, z) P(v \mid x_0, w, z) P(w \mid x_1, z) P(z) \tag{27}$$

$$= \text{Eq. (2)}. \tag{28}$$

Also,

$$\mathbb{E}[\pi_0^2(W, X, Z) \mu_0^2(W, X, Z)] = \sum_{w,x,z} \mu_0^2(w, x, z) \frac{P(w \mid x_1, z) 1[x = x_0]}{P(w \mid x, z) P(x \mid z)} P(w, x, z) \tag{29}$$

$$= \sum_{w,z} \mu_0^2(w, x_0, z) P(w \mid x_1, z) P(z) \tag{30}$$

$$= \sum_{w,z} \sum_v \mathbb{E}[Y \mid v, w, x_1, z] P(v \mid w, x_0, z) P(w \mid x_1, z) P(z) \tag{31}$$

$$= \text{Eq. (2)}. \tag{32}$$

Finally,

$$\mathbb{E}[\pi_0^1(X, Z) \mu_0^1(X, Z)] = \mathbb{E}[\mu_0^1(x_1, Z)] = \text{Eq. (2)}. \tag{33}$$

This completes the proof.

## A.4 Proof of Lemma 2

We will first use the following helper lemma:

**Lemma 3** (**Helper Lemma**). *For any functional* $(a_0(\mathbf{W}), b_0(\mathbf{W}))$ *and* $(a(\mathbf{W}), b(\mathbf{W}))$, *for any* $\mathbf{W} \subseteq \mathbf{V}$, *the following holds:*

$$\mathbb{E}[a(\mathbf{W})\{b_0(\mathbf{W}) - b(\mathbf{W})\} + a_0(\mathbf{W})b(\mathbf{W}) - a_0(\mathbf{W})b_0(\mathbf{W})] \tag{34}$$

$$= \mathbb{E}[\{a_0(\mathbf{W}) - a(\mathbf{W})\}\{b(\mathbf{W}) - b_0(\mathbf{W})\}]. \tag{35}$$

*Proof.* Define $F(\mathbf{W})$ as a function satisfying the following equation:

$$\mathbb{E}[a(\mathbf{W})b(\mathbf{W})] - \mathbb{E}[a_0(\mathbf{W})b_0(\mathbf{W})] = F(\mathbf{W}) + \mathbb{E}[\{a_0(\mathbf{W}) - a(\mathbf{W})\}\{b(\mathbf{W}) - b_0(\mathbf{W})\}]. \tag{36}$$

We will omit $\mathbf{W}$ for notational convenience for now. The above equation shows that

$$F = \mathbb{E}[ab - a_0 b_0 - (a_0 - a)(b - b_0)] \tag{37}$$

$$= \mathbb{E}[ab - a_0 b + a_0 b_0 + ab - ab_0 - a_0 b_0] \tag{38}$$

$$= \mathbb{E}[2ab - a_0 b - ab_0]. \tag{39}$$

Then,

$$\mathbb{E}[ab - F] = \mathbb{E}[a_0 b + ab_0 - ab] = \mathbb{E}[a(\mathbf{W})(b_0 - b) + a_0 b]. \tag{40}$$

By the definition of $F$, we have

$$\mathbb{E}[ab - F] - \mathbb{E}[a_0 b_0] = \mathbb{E}[\{a_0(\mathbf{W}) - a(\mathbf{W})\}\{b(\mathbf{W}) - b_0(\mathbf{W})\}]. \tag{41}$$

This completes the proof. $\qquad\square$

Based on this helper lemma, for $i = 1, 2, 3$, we have

$$\mathbb{E}[\pi^i\{\mu_0^i - \mu^i\} + \pi_0^i \mu^i - \pi_0^i \mu_0^i] = \mathbb{E}[\{\mu_0^i - \mu^i\}\{\pi^i - \pi_0^i\}]. \tag{42}$$

Then, consider

$$\mathbb{E}[\pi^3\{\mu_0^3 - \mu^3\} + \pi_0^3 \mu^3 - \pi_0^3 \mu_0^3] \tag{43}$$

$$= \mathbb{E}[\pi^3\{Y - \mu^3\} + \pi_0^3 \mu^3] - \psi_0 \tag{44}$$

$$= \mathbb{E}[\{\mu^3 - \mu_0^3\}\{\pi_0^3 - \pi^3\}], \tag{45}$$

where the equation holds since $\mathbb{E}[\pi_0^3 \mu_0^3] = \psi_0 \triangleq$ Eq. (2), and by the law of the total expectation.

Next, define $\mu_*^2 \triangleq \mu_*^2[\mu^3] \triangleq \mathbb{E}[\mu^3(V, W, x_1, Z) \mid W, X, Z]$ for any fixed $\mu^3$. Then,

$$\mathbb{E}[\pi^2\{\mu_*^2 - \mu^2\} + \pi_0^2 \mu^2 - \pi_0^2 \mu_*^2] \tag{46}$$

$$= \mathbb{E}[\pi^2\{\mu^3(V, W, x_1, Z) - \mu^2\} + \pi_0^2 \mu^2] - \mathbb{E}[\pi_0^2 \mu_*^2] \tag{47}$$

$$= \mathbb{E}[\{\mu^2 - \mu_*^2\}\{\pi_0^2 - \pi^2\}]. \tag{48}$$

Next, define $\mu_*^1 \triangleq \mu_*^1[\mu^2] \triangleq \mathbb{E}[\mu^1(W, x_0, Z) \mid X, Z]$ for any fixed $\mu^2$. Then,

$$\mathbb{E}[\pi^1\{\mu_*^1 - \mu^1\} + \pi_0^1 \mu^1 - \pi_0^1 \mu_*^1] \tag{49}$$

$$= \mathbb{E}[\pi^1\{\mu^2(V, W, x_1, Z) - \mu^1\} + \pi_0^1 \mu^1] - \mathbb{E}[\pi_0^1 \mu_*^1] \tag{50}$$

$$= \mathbb{E}[\{\mu^1 - \mu_*^1\}\{\pi_0^1 - \pi^1\}]. \tag{51}$$

Combining,

$$\mathbb{E}[\pi^3(V, W, X, Z)\{Y - \mu^3(V, W, X, Z)\} + \pi_0^3(V, W, X, Z)\mu^3(V, W, X, Z)] - \psi_0 \tag{52}$$

$$+ \mathbb{E}[\pi^2(W, X, Z)\{\mu^3(V, W, x_1, Z) - \mu^2(W, X, Z)\} + \pi_0^2(W, X, Z)\mu^2(W, X, Z)] - \mathbb{E}[\pi_0^2 \mu_*^2] \tag{53}$$

$$+ \mathbb{E}[\pi^1(X, Z)\{\mu^2(W, x_0, Z) - \mu^1(X, Z)\} + \pi_0^1(X, Z)\mu^1(X, Z)] - \mathbb{E}[\pi_0^1 \mu_*^1] \tag{54}$$

$$= \sum_{i=1}^{3} \mathbb{E}[\{\mu^i - \mu_0^i\}\{\pi_0^i - \pi^i\}]. \tag{55}$$

Then,

$$\mathbb{E}[\pi_0^3(V,W,X,Z)\mu^3(V,W,X,Z)] = \mathbb{E}[\pi_0^2(W,X,Z)\mu_*^2(W,X,Z)], \tag{56}$$

since

$$\mathbb{E}[\pi_0^2(W,X,Z)\mu_*^2(W,X,Z)] \tag{57}$$

$$= \mathbb{E}[\pi_0^2(W,X,Z)\mathbb{E}[\mu^3(V,W,x_1,Z) \mid W,X,Z]] \tag{58}$$

$$= \sum_{v,w,x,z} \mu^3(v,w,x_1,z)P(v \mid w,x,z)\pi_0^2(w,x,z)P(w,x,z) \tag{59}$$

$$= \sum_{v,w,x,z} \mu^3(v,w,x_1,z)P(v \mid w,x,z)\frac{P(w \mid x_1,z)1(x=x_0)}{P(w \mid x,z)P(x \mid z)}P(w,x,z) \tag{60}$$

$$= \sum_{v,w,z} \mu^3(v,w,x_1,z)P(v \mid w,x_0,z)P(w \mid x_1,z)P(z) \tag{61}$$

$$= \sum_{v,w,x,z} \mu^3(v,w,x,z)\frac{1(x=x_1)}{P(x \mid z)}\frac{P(v \mid w,x_0,z)}{P(v \mid w,x,z)}P(v \mid w,x,z)P(w \mid x,z)P(x \mid z)P(z) \tag{62}$$

$$= \mathbb{E}[\mu^3(V,W,X,Z)\pi_0^3(V,W,X,Z)]. \tag{63}$$

Therefore, $\mathbb{E}[\pi_0^3\mu^3]$ in the first term and $-\mathbb{E}[\pi_0^2\mu_*^2]$ in the second term can be canceled out.

Furthermore,

$$\mathbb{E}[\pi_0^2(W,X,Z)\mu^2(W,X,Z)] = \mathbb{E}[\pi_0^1(X,Z)\mu_*^1(X,Z)], \tag{64}$$

since

$$\mathbb{E}[\pi_0^1(X,Z)\mu_*^1(X,Z)] = \mathbb{E}[\mu_*^1(x_1,Z)] \tag{65}$$

$$= \mathbb{E}[\mathbb{E}[\mu^2(W,x_0,Z)] \mid x_1,Z] \tag{66}$$

$$= \sum_z \sum_w \mu^2(w,x_0,z)P(w \mid x_1,z)P(z) \tag{67}$$

$$= \sum_{w,x,z} \mu^2(w,x,z)\frac{1[x=x_0]P(w \mid x_1,z)}{P(w \mid x,z)P(x \mid z)}P(w,x,z) \tag{68}$$

$$= \mathbb{E}[\pi_0^2(W,X,Z)\mu^2(W,X,Z)]. \tag{69}$$

Therefore, $\mathbb{E}[\pi_0^2\mu^2]$ in the second term and $-\mathbb{E}[\pi_0^1\mu_*^1]$ in the third term can be canceled out. Therefore, we can conclude that

$$\mathbb{E}[\pi^3(V,W,X,Z)\{Y - \mu^3(V,W,X,Z)\} \tag{70}$$

$$+ \mathbb{E}[\pi^2(W,X,Z)\{\mu^3(V,W,x_1,Z) - \mu^2(W,X,Z)\} \tag{71}$$

$$+ \mathbb{E}[\pi^1(X,Z)\{\mu^2(W,x_1,Z) - \mu^1(X,Z) + \mu^1(x,Z)\} \tag{72}$$

$$- \psi_0 \tag{73}$$

$$= \sum_{i=1}^3 \mathbb{E}[\{\mu^i - \mu_0^i\}\{\pi_0^i - \pi^i\}]. \tag{74}$$

This completes the proof.

## A.5   Proof of Theorem 2

### A.5.1   Proof of Eq. (8) in Theorem 2

Let $\mathbf{V} = \{Y,V,W,X,Z\}$. We note that the doubly robust estimator is given as

$$\hat{\psi} = \frac{1}{L}\sum_{\ell=1}^L \mathbb{E}_{\mathcal{D}_\ell}[\varphi(\mathbf{V};\widehat{\boldsymbol{\mu}}_\ell,\widehat{\boldsymbol{\pi}}_\ell)]. \tag{75}$$

Then,

$$\hat{\psi} - \psi_0 \tag{76}$$

$$= \frac{1}{L} \sum_{\ell=1}^{L} \mathbb{E}_{\mathcal{D}_\ell}[\varphi(\mathbf{V}; \widehat{\boldsymbol{\mu}}_\ell, \widehat{\boldsymbol{\pi}}_\ell)] - \mathbb{E}_P[\varphi(\mathbf{V}; \boldsymbol{\mu}_0, \boldsymbol{\pi}_0)] \tag{77}$$

$$= \frac{1}{L} \sum_{\ell=1}^{L} (\mathbb{E}_{\mathcal{D}_\ell} - \mathbb{E}_P)[\varphi(\mathbf{V}; \boldsymbol{\mu}_0, \boldsymbol{\pi}_0)] \tag{78}$$

$$+ \frac{1}{L} \sum_{\ell=1}^{L} (\mathbb{E}_{\mathcal{D}_\ell} - \mathbb{E}_P)[\varphi(\mathbf{V}; \widehat{\boldsymbol{\mu}}, \widehat{\boldsymbol{\pi}}) - \varphi(\mathbf{V}; \boldsymbol{\mu}_0, \boldsymbol{\pi}_0)] \tag{79}$$

$$+ \frac{1}{L} \sum_{\ell=1}^{L} \mathbb{E}_P[[\varphi(\mathbf{V}; \widehat{\boldsymbol{\mu}}, \widehat{\boldsymbol{\pi}}) - \varphi(\mathbf{V}; \boldsymbol{\mu}_0, \boldsymbol{\pi}_0)]. \tag{80}$$

Define

$$R_1 \triangleq \text{Eq. (78)} + \text{Eq. (79)} = \frac{1}{L} \sum_{\ell=1}^{L} (\mathbb{E}_{\mathcal{D}_\ell} - \mathbb{E}_P)[\varphi(\mathbf{V}; \widehat{\boldsymbol{\mu}}, \widehat{\boldsymbol{\pi}})]. \tag{81}$$

Then, with the proof of Lemma 3, we observe that Eq. (8) holds.

### A.5.2 Proof of Eq. (9) in Theorem 2

Let

$$\phi_0(\mathbf{V}) \triangleq \phi(\mathbf{V}; \boldsymbol{\mu}_0, \boldsymbol{\pi}_0), \tag{82}$$

$$\widehat{\phi}_\ell(\mathbf{V}) \triangleq \phi(\mathbf{V}; \widehat{\boldsymbol{\mu}}_\ell, \widehat{\boldsymbol{\pi}}_\ell). \tag{83}$$

We first study the term $(\mathbb{E}_{\mathcal{D}} - \mathbb{E}_P)[\phi_0(\mathbf{V})]$. By Chebyshev's inequality,

$$P\left(\left|(\mathbb{E}_{\mathcal{D}} - \mathbb{E}_P)[\phi_0(\mathbf{V})]\right| > t_1 \frac{\rho_0^2}{\sqrt{n}}\right) < \frac{1}{t_1^2}, \tag{84}$$

where $n \triangleq |\mathcal{D}|$. Equivalently,

$$P\left(\left|(\mathbb{E}_{\mathcal{D}} - \mathbb{E}_P)[\phi_0(\mathbf{V})]\right| > t_1\right) < \frac{1}{t_1^2} \frac{\rho_0^2}{n}, \tag{85}$$

or equivalently,

$$P\left(\left|(\mathbb{E}_{\mathcal{D}} - \mathbb{E}_P)[\phi_0(\mathbf{V})]\right| \leq t_1\right) > 1 - \frac{1}{t_1^2} \frac{\rho_0^2}{n}. \tag{86}$$

By [Lemma D.3 in 25], we have

$$P\left(\left|(\mathbb{E}_{\mathcal{D}_\ell} - \mathbb{E}_P)[\widehat{\phi}_\ell(\mathbf{V}) - \phi_0(\mathbf{V})]\right| > t_2\right) < \frac{1}{t_2^2} \frac{L\|\widehat{\phi}_\ell(\mathbf{V}) - \phi_0(\mathbf{V})\|_P^2}{n}. \tag{87}$$

By [Lemma D.4 in 25], we have

$$P\left(\frac{1}{L} \sum_{\ell=1}^{L} \left|(\mathbb{E}_{\mathcal{D}_\ell} - \mathbb{E}_P)[\widehat{\phi}_\ell(\mathbf{V}) - \phi_0(\mathbf{V})]\right| \leq t_2\right) \geq 1 - \sum_{\ell=1}^{L} \frac{1}{t_2^2} \frac{L\|\widehat{\phi}_\ell(\mathbf{V}) - \phi_0(\mathbf{V})\|_P^2}{n}. \tag{88}$$

Choose

$$t_1 = \sqrt{\frac{\epsilon}{2} \frac{\rho_0^2}{n}}, \tag{89}$$

$$t_2 = \sqrt{\frac{\epsilon}{2} \sum_{\ell=1}^{L} \frac{L\|\widehat{\phi}_\ell(\mathbf{V}) - \phi_0(\mathbf{V})\|_P^2}{n}}. \tag{90}$$

Then, with a probability greater than $1 - \epsilon$,

$$R_1 \le t_1 + t_2 \tag{91}$$

$$= \sqrt{\frac{\epsilon}{2} \frac{\rho_0^2}{n}} + \sqrt{\frac{\epsilon}{2} \sum_{\ell=1}^{L} \frac{L\|\widehat{\phi}_\ell(\mathbf{V}) - \phi_0(\mathbf{V})\|_P^2}{n}} \tag{92}$$

$$= \sqrt{\frac{\epsilon}{2}} \left( \sqrt{\frac{\rho_0^2}{n}} + \sqrt{\sum_{\ell=1}^{L} \frac{L\|\widehat{\phi}_\ell(\mathbf{V}) - \phi_0(\mathbf{V})\|_P^2}{n}} \right). \tag{93}$$

### A.5.3   Proof of Eq. (10) in Theorem 2

From the previous proof, we have

$$P\left( \frac{1}{L} \sum_{\ell=1}^{L} \left| (\mathbb{E}_{\mathcal{D}_\ell} - \mathbb{E}_P)[\widehat{\phi}_\ell(\mathbf{V}) - \phi_0(\mathbf{V})] \right| \le t_2 \right) \ge 1 - \sum_{\ell=1}^{L} \frac{1}{t_2^2} \frac{L\|\widehat{\phi}_\ell(\mathbf{V}) - \phi_0(\mathbf{V})\|_P^2}{n}. \tag{94}$$

By choosing

$$t_2 = \sqrt{\frac{1}{\epsilon} \sum_{\ell=1}^{L} \frac{L\|\widehat{\phi}_\ell(\mathbf{V}) - \phi_0(\mathbf{V})\|_P^2}{n}}, \tag{95}$$

we have

$$\frac{1}{L} \sum_{\ell=1}^{L} \left| (\mathbb{E}_{\mathcal{D}_\ell} - \mathbb{E}_P)[\widehat{\phi}_\ell(\mathbf{V}) - \phi_0(\mathbf{V})] \right| \overset{\text{with probability } 1 - \epsilon}{\le} \sqrt{\frac{1}{\epsilon} \sum_{\ell=1}^{L} \frac{L\|\widehat{\phi}_\ell(\mathbf{V}) - \phi_0(\mathbf{V})\|_P^2}{n}}. \tag{96}$$

Define

$$A \triangleq (\mathbb{E}_{\mathcal{D}} - \mathbb{E}_P)[\phi_0(\mathbf{V})], \tag{97}$$

$$B \triangleq \frac{1}{L} \sum_{\ell=1}^{L} (\mathbb{E}_{\mathcal{D}_\ell} - \mathbb{E}_P)[\widehat{\phi}_\ell(\mathbf{V}) - \phi_0(\mathbf{V})], \tag{98}$$

$$C \triangleq \frac{1}{L} \sum_{\ell=1}^{L} \left| (\mathbb{E}_{\mathcal{D}_\ell} - \mathbb{E}_P)[\widehat{\phi}_\ell(\mathbf{V}) - \phi_0(\mathbf{V})] \right|, \tag{99}$$

$$\Delta \triangleq \sqrt{\frac{1}{\epsilon} \sum_{\ell=1}^{L} \frac{L\|\widehat{\phi}_\ell(\mathbf{V}) - \phi_0(\mathbf{V})\|_P^2}{n}}. \tag{100}$$

Here,

$$R_1 = A + B. \tag{101}$$

Then,

$$P(R < x) = P(A + B < x) = P(A < x - B) \le P(A < x + C) \overset{\text{w.p. } 1 - \epsilon}{\le} P(A < x + \Delta). \tag{102}$$

Then,

$$\left| P(A < x + \Delta) - \Phi(x) \right| \tag{103}$$

$$= \left| P(A < x + \Delta) - \Phi(x + \Delta) + \Phi(x + \Delta) - \Phi(x) \right| \tag{104}$$

$$\le \left| P(A < x + \Delta) - \Phi(x + \Delta) \right| + \left| \Phi(x + \Delta) - \Phi(x) \right| \tag{105}$$

$$\le \frac{0.4748\kappa_0^3}{\rho_0^3 \sqrt{n}} + \left| \Phi(x + \Delta) - \Phi(x) \right| \quad \text{[Prop. D.1 in 25]} \tag{106}$$

$$= \frac{0.4748\kappa_0^3}{\rho_0^3 \sqrt{n}} + \left| \Phi'(x')\Delta \right| \quad \text{(mean-value theorem)} \tag{107}$$

$$\le \frac{0.4748\kappa_0^3}{\rho_0^3 \sqrt{n}} + \frac{1}{\sqrt{2\pi}}\Delta. \tag{108}$$

This completes the proof.

## B  Self-normalized estimators for the NDE and NIE

We focus on estimating the counterfactual term for the NDE and NIE, $\mathbb{E}[Y_{x_1, W_{x_0}, V_{x_0}}]$, since estimating $\mathbb{E}[Y_{x_0}]$ and $\mathbb{E}[Y_{x_1}]$ using the back-door adjustment [35] is well-known. Let $M = \{W, V\}$ be the combined set of mediators. The counterfactual quantity is identifiable in Figure 2 using the formula:

$$\mathbb{E}[Y_{x_1, M_{x_0}}] = \sum_{m,z} \mathbb{E}[Y \mid x_1, m, z] P(m \mid x_0, z) P(z).$$

Following the recipe in [25], the expression for $\psi_0 = \mathbb{E}[Y_{x_1, M_{x_0}}]$ can be parameterized using the following nuisance parameters:

| Regression Parameters $\mu_0$ | | Importance Sampling Parameters $\pi_0$ | |
|---|---|---|---|
| $\mu_0^2(M, X, Z)$ | $\triangleq \mathbb{E}[Y \mid M, X, Z]$ | $\pi_0^2(M, X, Z)$ | $\triangleq \frac{P(M \mid x_0, Z)}{P(M \mid X, Z)} \frac{1[X = x_1]}{P(X \mid Z)}$ |
| $\mu_0^1(X, Z)$ | $\triangleq \mathbb{E}[\breve{\mu}_0^2 \mid X, Z]$ | $\pi_0^1(X, Z)$ | $\triangleq \frac{1[X = x_0]}{P(X \mid Z)}$ |

where $\breve{\mu}_0^2 \triangleq \mu_0^2(M, x_1, Z)$. The doubly robust estimator for $\psi_0$ is

$$\hat{\psi} = \mathbb{E}[\hat{\pi}_0^2 \{Y - \hat{\mu}_0^2\}] + \mathbb{E}[\hat{\pi}_0^1 \{\breve{\mu}_0^2 - \hat{\mu}_0^1\}] + \mathbb{E}[\breve{\mu}_0^1].$$

We estimate the nuisance parameters using the same sample-splitting procedure as for the VDE. To estimate $\hat{\pi}^i \in \hat{\boldsymbol{\pi}}$, we can rewrite the expressions using Bayes' rule to avoid computing high-dimensional densities. Since $\mathbb{E}[\pi_0^i] = 1$, the self-normalized variant uses $\hat{\pi}_{\text{SN}}^i \leftarrow \hat{\pi}^i / \mathbb{E}_{\mathcal{D}}[\hat{\pi}^i]$.

## C  Additional data analysis

All analysis is restricted to adult patients. To better understand the distribution of patients in our datasets, we analyze baseline characteristics and pre-admission severity scores across datasets and racial groups. Figure 8 shows the distribution of age and sex, stratified by race. To assess pre-ICU severity, we use two measures: the OASIS score and the Charlson Comorbidity Index. Both are integer-based scores, with higher values indicating greater clinical severity. These distributions are shown in Figure 9.

We observe that the distribution of patient severity remains consistent across different racial groups. Moreover, baseline patient conditions tend to be more severe in the eICU dataset, which explains the longer ventilation times in Figure 4b.

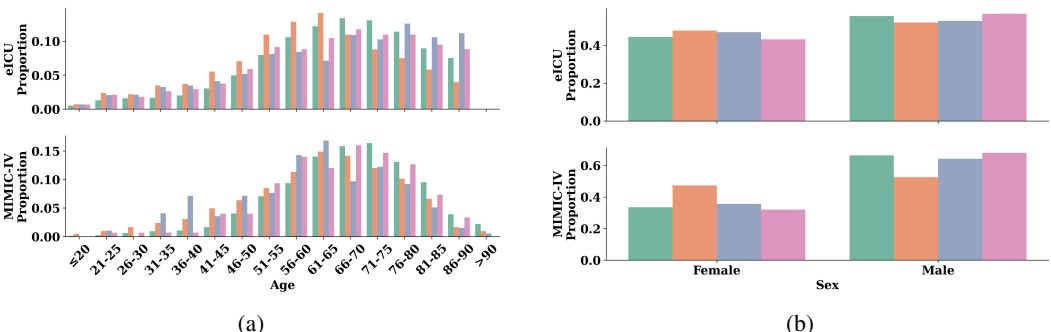

(a)                   (b)

**Figure 8.** Distribution of patient baseline characteristics, (a) age and (b) sex, in the eICU and MIMIC-IV datasets, stratified by race.

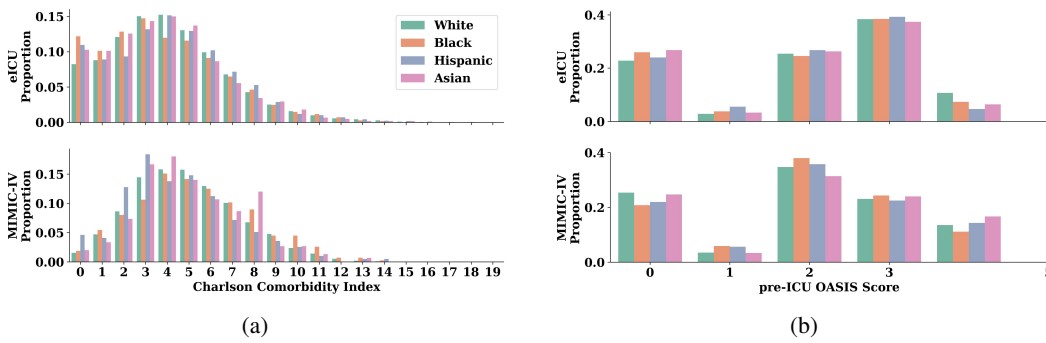

(a)                                       (b)

**Figure 9.** Distribution of pre-admission severity scores, (a) Charlson Comorbidity Index and (b) pre-ICU OASIS Score, in the eICU and MIMIC-IV datasets, stratified by race. The eICU cohort exhibits a slightly higher pre-admission severity score.

## D   Hyperparameter selection

In this section, we outline our approach for selecting the hyperparameters of the XGBoost models to compute $\mathbb{E}[Y_{x_1, W_{x_0}, V_{x_0}}]$ and $\mathbb{E}[Y_{x_1, V_{x_0}, W_{x_1}}]$. For both the synthetic and semi-synthetic settings, we generate a random dataset consisting of 32,000 samples and perform a separate grid search for each propensity model and each outcome model. For the real-world setting, we perform the hyperparameter search using the eICU and MIMIC-IV datasets.

The grid search optimizes three hyperparameters: the number of estimators from the set $\{20, 50, 100, 200\}$, the maximum tree depth from the set $\{3, 4, 5, 6\}$, and the $\ell_2$-regularization penalty from the set $\{0.5, 1, 2, 5\}$, using 5-fold cross-validation. For propensity models, we select the hyperparameter configuration according to the Brier score, which evaluates prediction calibration. For outcome models, we select the configuration using the mean squared error. The chosen hyperparameters remain fixed across all bootstraps and sample sizes throughout the corresponding experimental setting.

**Table 3.** Selected hyperparameters for each model used in the real-world ICU experiments. Each tuple represents the best (tree depth, number of estimators, $\ell_2$-regularization), respectively. For the nested regression models in the bottom four rows, we specify the computed effect in parenthesis to distinguish between regression parameters for different estimates.

| Model | Vent. Rate (eICU) | Vent. Rate (MIMIC-IV) | Vent. Dur. (eICU) | Vent. Dur. (MIMIC-IV) |
|---|---|---|---|---|
| $p(X \mid Z)$ | $(3, 20, 5)$ | $(3, 20, 5)$ | $(3, 20, 5)$ | $(3, 20, 1)$ |
| $p(X \mid W, Z)$ | $(3, 20, 5)$ | $(3, 20, 5)$ | $(3, 20, 5)$ | $(3, 20, 1)$ |
| $p(X \mid V, Z)$ | $(3, 20, 5)$ | $(3, 20, 5)$ | $(3, 20, 2)$ | $(3, 20, 2)$ |
| $p(X \mid W, V, Z)$ | $(3, 20, 5)$ | $(3, 20, 5)$ | $(3, 20, 5)$ | $(3, 20, 5)$ |
| $\mathbb{E}[Y \mid x_0, V, Z]$ | $(3, 50, 5)$ | $(3, 20, 5)$ | $(3, 20, 2)$ | $(3, 20, 5)$ |
| $\mathbb{E}[Y \mid x_1, V, Z]$ | $(3, 20, 2)$ | $(3, 20, 5)$ | $(3, 20, 5)$ | $(3, 20, 5)$ |
| $\mathbb{E}[Y \mid x_0, W, V, Z]$ | $(3, 20, 2)$ | $(3, 20, 5)$ | $(3, 20, 5)$ | $(4, 20, 5)$ |
| $\mathbb{E}[Y \mid x_1, W, V, Z]$ | $(3, 20, 5)$ | $(3, 20, 5)$ | $(3, 20, 2)$ | $(6, 20, 5)$ |
| $\mathbb{E}[\check{\mu}^3 \mid x_0, W, Z]$ (VDE) | $(4, 200, 5)$ | $(4, 200, 2)$ | $(3, 200, 1)$ | $(3, 200, 5)$ |
| $\mathbb{E}[\check{\mu}^2 \mid x_1, Z]$ (VDE) | $(3, 20, 5)$ | $(3, 20, 5)$ | $(3, 20, 5)$ | $(3, 20, 5)$ |
| $\mathbb{E}[\check{\mu}^2 \mid x_0, Z]$ (NDE/NIE) | $(4, 20, 5)$ | $(3, 20, 5)$ | $(3, 20, 2)$ | $(3, 20, 5)$ |
| $\mathbb{E}[\check{\mu}^2 \mid x_0, Z]$ (NIE*) | $(4, 20, 1)$ | $(3, 20, 5)$ | $(3, 20, 5)$ | $(3, 20, 5)$ |

# E  VDE nuisance parameter analysis

We demonstrate that the improved variance of our proposed self-normalized estimator, as shown in Figures 3, can be attributed to better nuisance estimates when the sample size is small. In Figure 10, we present the mean and 95% confidence interval across 100 bootstraps for the empirical average of the VDE nuisance parameters $\hat{\pi}^3$, $\hat{\pi}^2$, and $\hat{\pi}^1$.

As the number of samples increases, the empirical mean converges to the true value of one. For smaller sample sizes, the nuisance parameters tend to be large, which causes greater variance in the standard doubly robust estimator. In contrast, our self-normalized variant scales the nuisance parameters so that the empirical average is one, which reduces the variance in the estimation. In the semi-synthetic setting, the nuisance parameters exhibit slow convergence or a small bias due to propensity clipping.

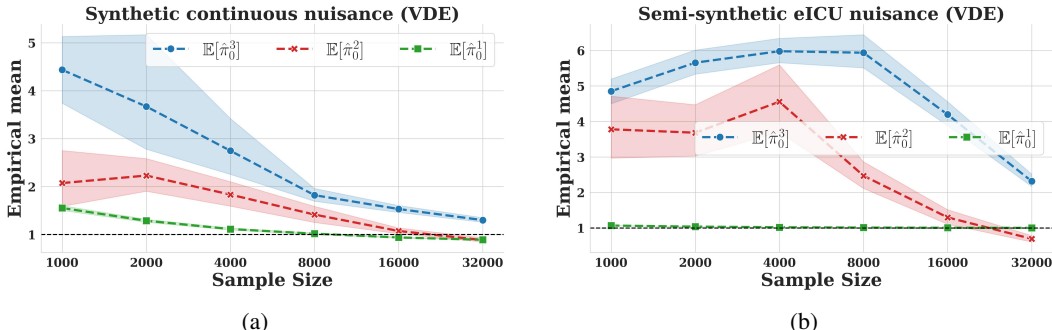

**Figure 10.** Convergence of the empirical mean of the VDE nuisance parameters in the (a) synthetic setting with continuous variables and the (b) semi-synthetic experiment based on eICU data. The dashed horizontal line indicates the theoretical mean for the nuisance parameters.

# F  Synthetic binary setting

In addition to the synthetic experiment with continuous variables, we also analyze a synthetic setting with singleton binary variables. This experiment aims to demonstrate the convergence of causal effects and nuisance parameter estimates. By choosing a simplified setting, we illustrate that when the nuisance parameters are accurate, the standard and self-normalized estimators produce approximately identical results.

We generate data according to the following model, where all observed variables take singleton binary values and the unobserved variables are $U_X, U_{XZ}, U_W, U_V, U_Y \sim \mathcal{N}(0, 1)$:

$$
\begin{aligned}
Z &= 1[U_{XZ} > 0.2], \\
X &= 1[Z + U_X + U_{XZ} > 0.2], \\
W &= 1[X - Z + U_W > 0.8], \\
V &= 1[X - Z + W + U_V > 0.8], \\
Y &= 1[X - Z + 2(W - V) + U_Y > 0.2].
\end{aligned}
$$

We estimate the causal queries necessary to compute fairness effects: $\mathbb{E}[Y_{x_0}]$, $\mathbb{E}[Y_{x_1}]$, $\mathbb{E}[Y_{x_1, M_{x_0}}]$, and $\mathbb{E}[Y_{x_1, V_{x_0}, W_{x_1}}]$. Like the synthetic setting with continuous variables, we vary the sample size from 1,000 to 32,000.

We report the mean and 95% confidence intervals across 100 bootstraps of the relative error on each causal query and the empirical mean of the nuisance parameters in Figure 11. As the sample size increases, our estimates converge to the true causal quantities. Due to the simplicity of the singleton binary variables in this experiment, the empirical mean of the nuisance parameters is always close to one. Consequently, the estimates obtained with the self-normalized estimator are nearly identical to those shown in Figure 11a.

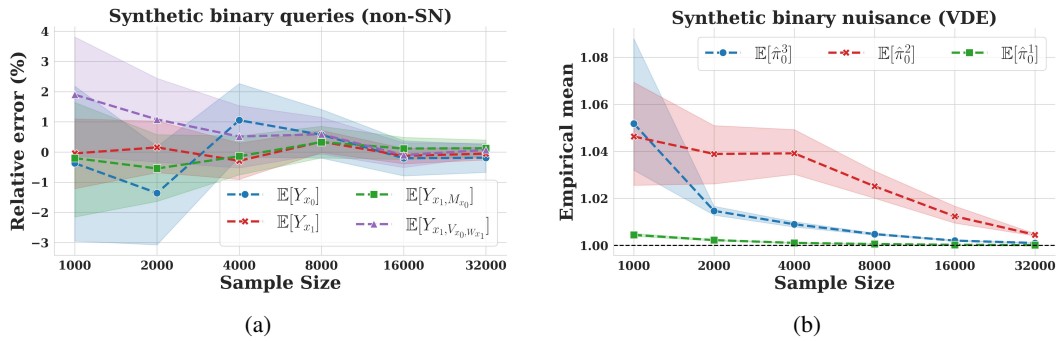

**Figure 11.** Convergence of (a) the relative error of causal queries using the canonical estimator and (b) the empirical mean of the nuisance parameters. Results for the self-normalized estimator are omitted as they are nearly identical to (a), because the empirical mean of the nuisance parameters is close to one.

# G   Semi-synthetic MIMIC-IV results

We show the experimental results for our estimators on semi-synthetic data based on patterns in the MIMIC-IV dataset. Following the setup in the semi-synthetic eICU setting, we train an XGBoost model to predict each variable in the set $\{X, W, V, Y\}$ given its observed parents in the real-world data. We fix $Y$ to be a binary variable indicating whether the patient received an invasive ventilation procedure during the stay.

We report the mean and 95% confidence interval across 100 bootstraps of the relative error for each causal query across varying sample sizes in Figure 12. Like for the semi-synthetic eICU setting, we observe that our estimands approximately converge to the true value, however, the convergence is slow for $\mathbb{E}[Y_{x_1, V_{x_0}, W_{x_1}}]$ using the canonical doubly robust estimator.

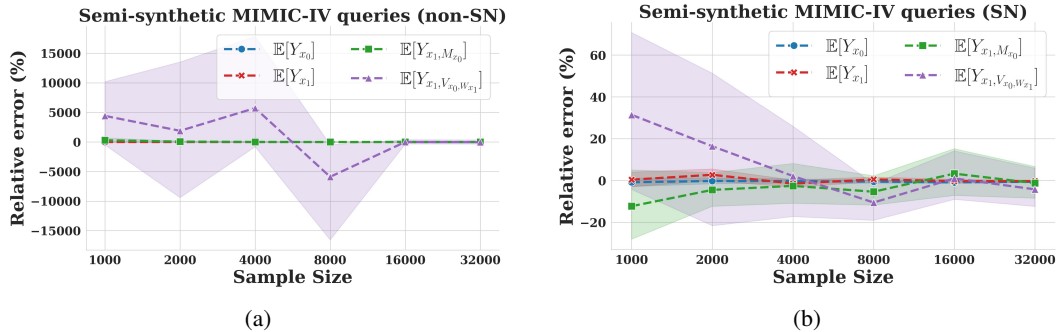

**Figure 12.** Convergence of the relative error of causal queries using the (a) canonical and (b) self-normalized estimator on the semi-synthetic MIMIC-IV data.

# H   Canonical doubly robust estimator results on real-world data

We compute the causal effects in Table 2 using the canonical doubly robust estimator. We report the mean and 95% confidence intervals for the rate and duration of invasive ventilation in Figure 13. The VDE for invasive ventilation rates for eICU ($-0.21$ percentage points, 95% CI $[-0.23$ to $-0.19]$) and MIMIC-IV ($-0.47$ percentage points, 95% CI $[-0.57$ to $-0.36]$) datasets is relatively small. Moreover, the VDE indicates slightly longer ventilation durations for Black relative to White patients in eICU data ($2.4$ hrs, 95% CI $[2.2$ to $2.5]$) and significantly longer durations in MIMIC-IV data ($8.1$ hrs, 95% CI $[7.5$ to $8.7]$).

The direction and statistical significance of all causal fairness measures obtained using the self-normalized estimator (Figure 5) and canonical estimator are in agreement. Furthermore, we observe that the canonical estimator typically predicts wider confidence intervals across all effects, which is consistent with our experimental results in the synthetic and semi-synthetic settings.

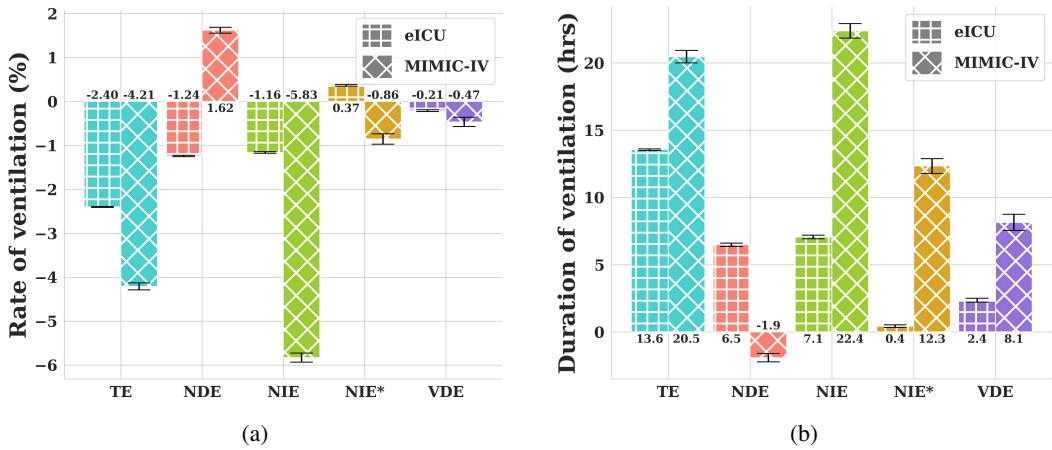

**Figure 13.** Average causal fairness measures across 500 bootstraps using the canonical doubly robust estimator for the (a) rate and (b) duration of invasive ventilation on eICU and MIMIC-IV data. Colors represent different effects and the numerical label for each bar indicates the mean across all bootstraps. Error bars show 95% confidence intervals. Positive values indicate a higher rate or duration of invasive ventilation for Black patients relative to White patients.

## I State-of-the-art comparison

We compare our proposed canonical doubly robust estimator and its self-normalized variant against several state-of-the-art and baseline methods. To the best of our knowledge, no prior work directly targets the VDE beyond the canonical estimator. Therefore, we include the following (somewhat misspecified) baselines: (1) the $X$-learner, which estimates the ATE without accounting for mediation; (2) a single-mediator estimator, which treats $W$ and $V$ as a single merged mediator (i.e., NIE* from Section 4.4); and (3) the nested regression estimator, corresponding to $\mu_0^1$ in Equation 6.

Table 4 presents the comparison of VDE prediction errors. Our results show that the self-normalized estimator consistently achieves the lowest error while maintaining valid 95% confidence intervals. Although the nested regression estimator performs similarly on semi-synthetic data, its confidence intervals are overly narrow. State-of-the-art methods for similar effect estimation tasks, such as the $X$-learner, have large error in all settings since they do not account for the correct mediation structure.

**Table 4.** Comparison of our canonical doubly robust estimator (Figures 3a, 3c) and self-normalized variant (Figures 3b, 3d) with state-of-the-art and baseline methods. We report the mean and standard error over 100 bootstraps, where each iteration samples a dataset of size 32,000. Relative errors are shown in parentheses, except for the semi-synthetic settings, where the true VDE is very close to zero and relative error is not informative. Entries in bold denote the best performing model for each dataset.

|  | Synthetic binary | Synthetic continuous | Semi-synthetic eICU | Semi-synthetic MIMIC |
|---|---|---|---|---|
| Ground truth VDE | $-0.125$ | $-0.163$ | $0.000$ | $0.001$ |
| $X$-learner | $0.364\,(292\%) \pm 0.001$ | $-0.352\,(217\%) \pm 0.001$ | $-0.007 \pm 0.000$ | $-0.029 \pm 0.000$ |
| Single-mediator doubly robust estimator [47, 49] | $-0.033\,(26\%) \pm 0.001$ | $-0.312\,(192\%) \pm 0.008$ | $0.024 \pm 0.014$ | $-0.027 \pm 0.004$ |
| Nested-regression estimator (Eq. 6) | $\mathbf{0.001\,(0.5\%) \pm 0.001}$ | $-0.024\,(15\%) \pm 0.002$ | $0.004 \pm 0.001$ | $\mathbf{-0.001 \pm 0.000}$ |
| Canonical doubly robust estimator [30] | $\mathbf{-0.001\,(0.5\%) \pm 0.001}$ | $\mathbf{0.006\,(4\%) \pm 0.008}$ | $0.018 \pm 0.020$ | $0.025 \pm 0.176$ |
| Self-normalized doubly robust estimator (ours) | $\mathbf{-0.001\,(0.5\%) \pm 0.001}$ | $\mathbf{0.006\,(4\%) \pm 0.006}$ | $\mathbf{0.003 \pm 0.007}$ | $0.004 \pm 0.004$ |

## J   Group imbalance experiments

In the real-world data, the eICU cohort comprises approximately 31,000 White and 4,000 Black patients, while MIMIC-IV includes around 4,000 White and 400 Black patients. These distributions reveal a substantial group imbalance between White and Black patients, which is approximately preserved in our semi-synthetic experiments.

Although our self-normalized estimator demonstrates strong convergence properties despite this imbalance, we further examine its performance under varying degrees of group imbalance. Using the eICU-based and MIMIC-based semi-synthetic datasets, we control the imbalance by thresholding the predictions of $X$ at different values to induce varying proportions of $x_0$ and $x_1$ subgroups. To quantify the imbalance, we define $\eta = \frac{\# \text{ of } x_1 \text{ samples}}{\# \text{ of total samples}}$. We assess performance using the relative error for each causal query in Figure 14.

As expected, both the relative error and variance of the estimators tend to increase as the imbalance between the $x_0$ and $x_1$ subgroups grows. Nevertheless, the overall variation in estimator performance across different values of $\eta$ remains modest, with relative errors increasing by only a few percentage points. These findings suggest that our estimator is moderately robust to the group imbalance present in the real-world experiments.

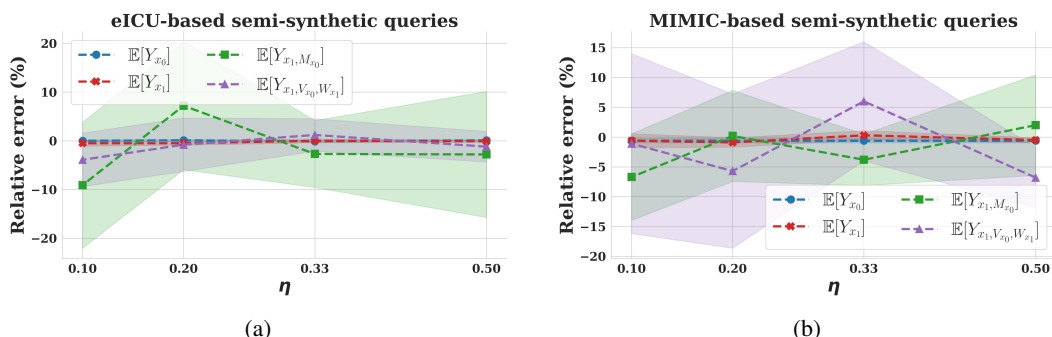

**Figure 14.** Causal fairness estimates for (a) eICU-based and (b) MIMIC-based semi-synthetic datasets, computed using our self-normalized estimator across varying levels of group imbalance ($\eta \in [0.1, 0.2, 0.33, 0.5]$). Each point represents the mean estimate with a 95% confidence interval computed over 100 bootstrap iterations, each sampling a dataset of size 32,000.

## K   Compute resources

All models were trained on a single NVIDIA RTX A6000 GPU, 32 CPU cores, 256GB of system RAM, running Ubuntu 22.04 with kernel 5.15. The synthetic and semi-synthetic experiments compute 100 bootstraps for all sample sizes in two hours, while the real-world experiments require two hours to compute 500 bootstraps for each dataset and ventilation-based outcome.

## L   Experiment assets

In our experiments, we use the eICU [40] v2.0 and MIMIC-IV [22] v3.1 datasets, both publicly available under the PhysioNet Credentialed Health Data License 1.5.0. Data processing is performed using the eICU-CRD Code Repository[4] and the MIMIC Code Repository[5], both under the MIT License. For eICU ventilation outcomes, we follow the data extraction procedure described in [48]. For Charlson Comorbidity Scores in the eICU dataset, we follow the process in [3] provided in the code implementation under the MIT License.[6] All the code and instructions for reproducing our experiments are available at `https://github.com/reAIM-Lab/PSE-Pulse-Oximetry`.

---

[4]`https://github.com/MIT-LCP/eicu-code`
[5]`https://github.com/MIT-LCP/mimic-code`
[6]`https://github.com/theonesp/vol_leak_index`

