# OpenReview forum: "Path-specific effects for pulse-oximetry guided decisions in critical care"
_NeurIPS.cc/2025/Conference — NeurIPS 2025 poster_

### Official Review · Reviewer_vpLv · 2025-06-25

**Clarity:** 2
**Significance:** 3
**Originality:** 3
**Rating:** 4
**Confidence:** 5

**Summary:**

This paper considers the fairness issues in Healthcare scenarios and aims for investigating the influence of racial bias in oximetry measurements on invasive ventilation in Intensive Care Units (ICU) settings from a causal perspective. Specifically, it leverages path-specific analysis to detect the causal effects of sensitive attribute (race) on invasive ventilation mediated by pulse oximetry discrepancies. To quantify such path-specific effects, it proposes a doubly robust estimator and a self-normalized variant, with fine-sample guarantees. Experiments on synthetic and real-world datasets showcase the effectiveness of the proposed method.

**Questions:**

1. What are the unique challenges of the ICU settings compared to other healthcare applications scenarios?
2. Please refer to the weaknesses.

**Ethical Concerns:**

["NO or VERY MINOR ethics concerns only"]

**Final Justification:**

The authors have addressed most of my questions, and I‘ll keep my rating.

**Limitations:**

yes

**Quality:**

3

**Strengths And Weaknesses:**

Strengths:
1. The authors address a nice and importance fairness problem in this work.
2. The authors propose a novel doubly robust estimator and self-normalized variant for quantifying V-specific direct effects.
3. Theoretically sound. The proposed method is built upon existing theoretical results while also providing rigorous theoretical analysis.
4. The experiments results demonstrate that the proposed method can effectively assess V-specific direct effects.

Weaknesses:
1. The presentation in articulating the research gap is not so clear. More explanations about how existing path-specific methods can be adapted or applied to ICU settings would help in understanding the flexibility and integration potential of them, e.g., Reference [27].
2. In Table 1, Reference [46] also focuses on multiple mediators settings.
3. The proposed V-specific direct effect is a special case of path-specific effects.
4. The validity of the proposed method relies on the assumptions of the causal models illustrated in Figure 2. However, the rationale behind data generation mechanism aligning with these causal models is not clearly presented. For example, X, representing race as an inherent characteristic of individuals, raises questions regarding why it is caused by Z. The authors do not provide any explanations where these causal models may hold, thus making it difficult to justify the assumptions.
5. There is a lack of comparison with SOTA methods in the experiments. More SOTA methods should be considered.

---

> ### Author Rebuttal · Authors · 2025-07-31
>
> Thank you for your detailed review and valuable suggestions. We address each of your comments below.
>
> **Weaknesses**
>
> 1. We will address weaknesses 1 and 2 together.
>
>    ###
>
>    In Table 1, [27], [44], and [46] investigate path-specific effects, but only [27] focus on effects isolated through a single mediator (as shown in the fourth column). [27] is conceptually similar to our non-self-normalized estimator which results in high variance (see Figures 3a and 3c), especially in small sample regimes like our application. [44] and [46] assume a single mediator, which could be applicable in other ICU settings, but is limiting in the pulse oximetry setting because many variables beyond the supplemental oxygenation discrepancy should be considered for valid fairness estimation. Nonetheless, we compare with this "misspecified" setting (i.e. one mediator), called the NIE$^*$, in Figure 5. We will add a discussion to address the research gap in the revisions.
>
> ###
>
> 4. As with most causal inference methods, our framework relies on a specified causal graph and thus makes assumptions about the data-generating process that can be challenging to verify in practice. To support these assumptions, we will expand the appendix to include a detailed discussion of their plausibility in the context of our ICU application. We will also emphasize that, although the validity of our methodology does indeed depend on the specified causal diagrams, using DAGs is advantageous because they clearly and explicitly represent the assumptions being made, and its importance has been highlighted in critical care literature as well [1]. These assumptions can be more easily verified or corrected compared to those in more opaque methods, making causal models relevant in medical (and other) domains.
>
>    ###
>
>    We agree that the variable $X$, representing race, should not be modeled as receiving a directed arrow from $Z$ alone. The more appropriate structure includes a hidden confounder between $X$ and $Z$, as demographic variables such as age may influence the distribution of racial groups. Importantly, introducing a bidirected arrow between $X$ and $Z$ does not affect our estimators or the theoretical analysis presented in Appendix A. We will revise the causal graph accordingly in the updated manuscript.
>
> ###
>
> 5. We agree with the reviewer and have included comparisons with three additional baseline methods in both the synthetic and semi-synthetic experiments. To the best of our knowledge, no existing work directly estimates the VDE beyond the canonical doubly robust estimator, which is challenging to interpret due to high variance. Therefore, we evaluate several state-of-the-art methods for similar effect estimation tasks. For details on the experimental setup and evaluation, please see our response 3 to Reviewer ScmX.
>
> **Questions**
> 1. The ICU is a particularly complex clinical setting because of its longitudinal nature and the high dimensionality of patient data. These characteristics make it difficult to apply standard algorithmic fairness frameworks directly. This motivates us to incorporate domain knowledge and a clinically grounded data-generating process to more appropriately assess fairness in this setting.
>
> We hope our response resolves your concerns and welcome any further questions you may have.
>
> [1] - David J. Lederer, Scott C. Bell, Richard D. Branson, et al. Control of confounding and reporting of results in causal inference studies. Guidance for authors from editors of respiratory, sleep, and critical care journals. *Annals of the American Thoracic Society*, 16, 2019.

---

> > ### Comment · Reviewer_vpLv · 2025-08-06
> >
> > The authors have addressed most of my questions. I will keep my score.

---

> ### Comment · Area_Chair_em4o · 2025-08-05
> **Please discuss your review**
>
> Hello reviewer vpLv - thank you again for taking the time to review this paper. A mandatory acknowledgement should only be done after you have had some discussion. Please comment on the paper to initiate the discussion. -Area Chair

---

### Official Review · Reviewer_ScmX · 2025-06-29

**Clarity:** 3
**Significance:** 3
**Originality:** 3
**Rating:** 4
**Confidence:** 4

**Summary:**

In this paper, the authors focus on tackling the problem of racial bias on ICU settings, where medical device bias can lead to delaying clinical interventions for certain sensitive groups. To this end, the authors introduce the V-specific direct effect to estimate the racial biases in the ICU. Subsequently, the authors propose a doubly robust estimator and a self-normalized variant for the V-specific direct effect with theoretical guarantees. Experimental results show the effectiveness of the proposed method in analyzing the causal effects of pulse oximeter mediated bias on invasive ventilation.

**Questions:**

Please refer to the weaknesses.

**Ethical Concerns:**

["NO or VERY MINOR ethics concerns only"]

**Final Justification:**

The authors have addressed most of my concerns, and I maintain my rating.

**Limitations:**

Yes, the authors have discussed the limitations of the proposed method.

**Quality:**

2

**Strengths And Weaknesses:**

Strengths:
1. The first causal analysis method to investigate the racial biases in ICU settings.
2. The authors propose a theoretically sound estimator to quantify the path-specific causal effects of racial bias on invasive ventilation.
3. The writing is clear and easy to follow.

Weaknesses:
1. The unique challenges of addressing fairness issues in ICU settings are not sufficiently discussed. In addition, it is unclear why existing causal analysis methods, such as [27], [44], and [46] shown in Table 1, are not applicable to ICU data.
2. The proposed method relies on several implicit assumptions to ensure the identifiability of the V-specific direct effect, e.g., the absence of hidden confounders. However, the plausibility of these assumptions in real ICU settings is not sufficiently discussed.
3. The numerical experiments lack comparisons with state-of-the-art methods, including those based on statistical correlations, to demonstrate the advantages of the proposed approach.
4. How does the proposed method performs under imbalanced sensitive group distributions, as is often the case in practice where black patients are underrepresented compared to white patients.

---

> ### Author Rebuttal · Authors · 2025-07-31
>
> Thank you for your thoughtful and constructive questions. We address all your concerns in the following response.
>
> **Weaknesses**
> 1. In Table 1, [27], [44], and [46] investigate path-specific effects, but only [27] focus on effects isolated through a single mediator (as shown in the fourth column). [27] is conceptually similar to our non-self-normalized estimator which results in high variance (see Figures 3a and 3c), especially in small sample regimes like our application. [44] and [46] allow only a single mediator, which could be applicable in other ICU settings, but is limited in the pulse oximetry setting because all relevant patient variables  beyond the supplemental oxygenation discrepancy should be considered for valid and unbiased fairness estimation. Nonetheless, we compare with this "misspecified" setting (i.e. assume only one mediator), called the NIE$^*$, in Figure 5. We will add a discussion to address the research gap in the revisions.
>
> ###
>
> 2. In Appendix A, we explicitly outline the identifiability conditions, which assume no unmeasured confounding between W and V in order to identify the VDE. This is reasonable in the context of our ICU application, and we will expand the appendix to include a discussion of the plausibility of this assumption based on the available covariates and the clinical context of our ICU application.
>
> ###
>
> 3. We thank the reviewer for identifying this point. To address the concern, we compare our proposed estimators, the canonical doubly robust estimator for the VDE and its self-normalized variant, with three additional baselines across synthetic (binary and continuous) and semi-synthetic (eICU and MIMIC-based) datasets. Since, to the best of our knowledge, no prior work directly targets the VDE beyond the canonical estimator, we include the following (somewhat misspecified) baselines:
>     - **X-learner**, which estimates the ATE without accounting for the mediation structure (ignoring $W$ and $V$), and cannot isolate the target effect we want.
>     - **Single-mediator doubly robust estimator**, which computes the effect through the mediators as if $W$ and $V$ were merged into a single variable (this is equivalent to the NIE$^*$ used in our real-world experiments).
>     - **Nested regression estimator**, which corresponds to $\mu_0^1$ in Equation 6.
>
>    We compare all five estimators in the table below, reporting the mean and standard error over 100 bootstraps, where each iteration samples a dataset of size 32,000. Relative errors are shown in parentheses, except for the semi-synthetic eICU setting, where the true VDE is very close to zero and relative error is not informative. Entries in bold denote the best performing model for each dataset.
>
>     |                                  | Synthetic binary                          | Synthetic continuous                       | Semi-synthetic eICU                | Semi-synthetic MIMIC                   |
>     |:---------------------------------|:-----------------------------------------:|:------------------------------------------:|:----------------------------------:|:-------------------------------------------:|
>     | **Ground truth**                | $-0.125\phantom{+}$                                  | $-0.163\phantom{+}$                                   | $0.000$                            | $0.001$                                     |
>     | **X-learner**                   | $\phantom{+}0.364\ (292\\\%) \pm 0.001$              | $-0.352\ (217\\\%) \pm 0.001$              | $-0.007 \pm 0.000$                | $-0.029\ (4819\\\%) \pm 0.000$              |
>     | **Single-mediator doubly robust estimator [44, 46]** | $-0.033\ (26\\\%) \pm 0.001$ | $-0.312\ (192\\\%) \pm 0.008$             | $\phantom{+}0.024 \pm 0.014$                 | $-0.027\ (4354\\\%) \pm 0.004$              |
>     | **Nested-regression estimator (Eq. 6, $\mu_0^1(x_1, Z)$ term)** | $\mathbf{\phantom{+}0.001\ (0.5\\\%) \pm 0.001}$ | $-0.024\ (15\\\%) \pm 0.002$         | $\phantom{+}0.004 \pm 0.001$                 | $\mathbf{-0.001\ (117\\\%) \pm 0.000}$           |
>     | **Canonical doubly robust estimator ([27], Fig. 3a&c)** | $\mathbf{-0.001\ (0.5\\\%) \pm 0.001}$ | $\mathbf{\phantom{+}0.006\ (4\\\%) \pm 0.008}$      | $\phantom{+}0.018 \pm 0.020$                 | $\phantom{+}0.025\ (4039\\\%) \pm 0.176$              |
>     | **Self-normalized doubly robust estimator (Proposed, Fig. 3b&d)** | $\mathbf{-0.001\ (0.5\\\%) \pm 0.001}$ | $\mathbf{\phantom{+}0.006\ (4\\\%) \pm 0.006}$      | $\mathbf{\phantom{+}0.003 \pm 0.007}$             | $\phantom{+}0.004\ (622\\\%) \pm 0.004$
>
>    The findings show that the self-normalized estimator typically yields the lowest error and maintains a $95\\%$ confidence interval that includes the true value. While the nested regression estimator performs comparably on semi-synthetic data, its confidence intervals are overly narrow. State-of-the-art methods for similar effect estimation tasks (i.e. the X-learner and single-mediator estimator) exhibit high error across all settings since they do not account for the correct mediation structure.
>
> ###
>
> 4. We have roughly 31,000 White and 4,000 Black patients in eICU, and around 4,000 White and 400 Black patients in MIMIC, thus capturing the imbalance you have correctly highlighted. This mainly reflects in the variance of the estimator, which we propose to mitigate using the proposed self-normalized variant. We also approximately preserve the imbalance in our semi-synthetic setup (the ratio of White to Black patients is 2:1). Figures 3c and 3d demonstrate that the self-normalized doubly robust estimator demonstrates good convergence behavior despite the imbalance.
>
>    ###
>
>    We also include an experiment demonstrating how varying levels of imbalance affect the performance of our estimator. Using the two semi-synthetic datasets, we threshold the predictions of $X$ at different values to induce varying proportions of $x_0$​ and $x_1$ subgroups. To quantify this imbalance, we define $\eta = \frac{\\# \text{ of $x_1$ samples}}{\\# \text{ of total samples}}$. Following the approach in Figure 3, we evaluate performance using the relative error for each causal query. The mean and standard error are reported over 100 bootstrap iterations, each sampling a dataset of size 32,000. Results are presented in the two tables below.
>
>    | **eICU-based semi-synthetic data**      | $\quad\quad \eta \approx 0.5$               | $\quad\\:\\:\\: \eta \approx 0.33$              | $\quad\quad \eta \approx 0.2$               | $\quad\quad \eta \approx 0.1$               |
>    |:----------------------------------------|:--------------------------------:|:--------------------------------:|:--------------------------------:|:--------------------------------:|
>    | $\mathbb{E}[Y_{x_0}]$                   | $\phantom{+}0.1\\% \pm 0.1\\%$   | $-0.1\\% \pm 0.1\\%$             | $\phantom{+}0.1\\% \pm 0.1\\%$   | $\phantom{+}0.0\\% \pm 0.1\\%$   |
>    | $\mathbb{E}[Y_{x_1}]$                   | $-0.1\\% \pm 0.1\\%$             | $\phantom{+}0.0\\% \pm 0.2\\%$   | $-0.5\\% \pm 0.2\\%$                | $-0.5\\% \pm 0.3\\%$             |
>    | $\mathbb{E}[Y_{x_1, M_{x_0}}]$          | $-2.8\\% \pm 6.6\\%$             | $-2.7\\% \pm 3.5\\%$             | $\phantom{+}7.2\\% \pm 6.7\\%$   | $-9.1\\% \pm 6.6\\%$             |
>    | $\mathbb{E}[Y_{x_1, V_{x_0}, W_{x_1}}]$ | $-1.2\\% \pm 1.6\\%$             | $\phantom{+}1.2\\% \pm 1.7\\%$   | $-0.8\\% \pm 2.8\\%$             | $-3.9\\% \pm 2.8\\%$             |
>
>    | **MIMIC-based semi-synthetic data** | $\quad\quad \eta \approx 0.5$             | $\quad\\:\\:\\: \eta \approx 0.33$            | $\quad\quad\eta \approx 0.2$             | $\quad\quad\eta \approx 0.1$             |
>    |:----------------------------------------|:------------------------------:|:------------------------------:|:------------------------------:|:------------------------------:|
>    | $\mathbb{E}[Y_{x_0}]$                   | $-0.6\\% \pm 0.2\\%$           | $-0.6\\% \pm 0.2\\%$           | $-0.6\\% \pm 0.2\\%$           | $-0.6\\% \pm 0.2\\%$           |
>    | $\mathbb{E}[Y_{x_1}]$                   | $-0.5\\% \pm 0.2\\%$           | $\phantom{+}0.3\\% \pm 0.4\\%$ | $-0.9\\% \pm 0.4\\%$           | $-0.6\\% \pm 0.6\\%$           |
>    | $\mathbb{E}[Y_{x_1, M_{x_0}}]$          | $\phantom{+}2.0\\% \pm 4.3\\%$ | $-3.8\\% \pm 2.2\\%$           | $\phantom{+}0.2\\% \pm 3.9\\%$ | $-6.7\\% \pm 3.7\\%$           |
>    | $\mathbb{E}[Y_{x_1, V_{x_0}, W_{x_1}}]$ | $-6.8\\% \pm 2.6\\%$           | $\phantom{+}6.0\\% \pm 5.1\\%$ | $-5.7\\% \pm 6.6\\%$           | $-1.1\\% \pm 7.7\\%$           |
>
>    As expected, the relative error and variance of the estimator generally increase with greater imbalance between the $x_0$ and $x_1$ subgroups (with the exception of $\mathbb{E}[Y_{x_1, V_{x_0, W_{x_1}}}]$ on MIMIC-based data). Nonetheless, the overall differences in estimator performance across levels of $\eta$ remain small, with relative error increasing by at most a few percentage points.
>
> We hope that our response addresses your concerns. We are happy to clarify any other questions you may have.

---

> > ### Comment · Reviewer_ScmX · 2025-08-07
> >
> > Thanks to the authors for the rebuttal. I think the author's comments have addressed most of my concerns, and suggest providing a more thorough explanation of the underlying assumptions in the revision to further strengthen the manuscript. I will maintain my score.

---

### Official Review · Reviewer_UBhm · 2025-07-02

**Clarity:** 3
**Significance:** 3
**Originality:** 3
**Rating:** 4
**Confidence:** 2

**Summary:**

The authors employ a causal inference framework to isolate the specific impact of pulse oximeter measurement discrepancies on invasive ventilation rates and duration. As key contributions, the authors propose a method using path-specific effects to quantify racial disparities in ICU decision-making. They define a specific causal query, the V-specific Direct Effect (VDE), to isolate the influence of race that is mediated specifically through oxygen saturation measurement discrepancies. They also present a doubly robust estimator for the VDE and introduces a self-normalized variant to improve sample efficiency and reduce variance, providing novel finite-sample guarantees for the estimator.

The methodology is validated on semi-synthetic data and then applied to two large ICU datasets, MIMIC-IV and eICU. Contrary to prior work suggesting significant disparities, this causal analysis finds a minimal impact of the measurement bias on invasive ventilation rates. However, the study does reveal a more pronounced and varied effect on the duration of ventilation, highlighting different outcomes between the two datasets.

**Questions:**

The paper finds a negligible effect of oximetry bias on the  rate of ventilation, which seems counterintuitive. Does this finding imply that for the critical decision of initiating ventilation, clinicians are successfully integrating a wide array of data beyond the pulse oximeter, effectively mitigating the device's known racial bias? Or could the bias be manifesting in ways not captured by this outcome, such as delays in receiving non-invasive oxygen support before a decision to intubate is made?

**Ethical Concerns:**

["NO or VERY MINOR ethics concerns only"]

**Final Justification:**

The authors promise expanded discussion. Maintain original assessment.

**Limitations:**

Yes.

**Quality:**

4

**Strengths And Weaknesses:**

Strengths
Novel causal inference framework applied to real-world dataset with insightful analysis.

Weakness
The authors acknowledge as a limitation that they only consider the first invasive ventilation and do not explicitly model temporality. They aggregate time-series data using an exponentially weighted average.

---

> ### Author Rebuttal · Authors · 2025-07-31
>
> Thank you for acknowledging the novelty of the method, and for your constructive comments and review.
>
> **Questions**
> 1. Thank you for the interesting question. We focus on the specific research question of path-specific effects of rate and duration of ventilation. We agree that we cannot rule out either of the two possibilities, that either the clinicians successfully integrate other information, reducing the overall effect of pulse oximeter discrepancy, or that other outcomes need to be measured, which is an active area of research. Our study is designed in line with the outcomes chosen in prior studies. We will add these details to our discussion.
>
> Please let us know if you have any additional questions and we are happy to clarify further.

---

### Official Review · Reviewer_1gtf · 2025-07-04

**Clarity:** 4
**Significance:** 3
**Originality:** 4
**Rating:** 5
**Confidence:** 4

**Summary:**

The manuscript describes a methodology to estimate path-specific causal effects in the presence of multiple mediators, with application to a fairness analysis in ICU data. In terms of the methodology, the authors develop a doubly robust estimator for the path-specific effect of a treatment on the outcome via one particular mediator. They provide a theoretical analysis of the properties of this estimator (identifiability, double robustness property, finite-sample behavior, asymptotic behavior) and verify its behavior in synthetic as well as semi-synthetic experiments. They then proceed to apply this estimator to the task of assessing whether measurement biases in pulse oximeters (which are known to produce mistakenly low measurements for Black patients at a higher rate than for white patients) have a specific effect on race-specific ventilation-related outcomes. This study is conducted in two ICU datasets, eICU and MIMIC-IV. The results indicate that following their methodology, the effect of these race-specific measurement discrepancies on ventilation outcomes appears to be small.

**Questions:**

- One thing left me a bit puzzled, but maybe I just missed a discussion of this somewhere? The authors estimate the V-specific effect, but V includes SpO2, SaO2, and the discrepancy between the two. In terms of the fairness question the authors are posing, it seems to me to make a crucial difference whether the authors are estimating an effect race -> SaO2 -> ventilation, or rather race -> (SpO2-SaO2) -> ventilation. The latter path would be unfair (and what the authors are really after, I believe) while the former path might be considered acceptable. Could the authors clarify this? E.g., which of these effects are the "VDEs" in Fig. 5?
- Possibly related to my previous question: in the same figure (5b), it appears highly unlikely that the effect of the measurement discrepancy would be positive in one study and negative in another. Is this because - as indicated in my previous question - the VDE here is actually *not* just the effect of the measurement discrepancy? In any case, the result as presented here would seem to indicate that the uncertainty estimation probably yields highly optimistic bounds, no? (Assuming that the true effect of measurement discrepancies should have the same directionality in both studies, the confidence intervals should at least cover the null in both studies.)
- In Fig. 3 top right panel, the variance actually does not appear to converge to 0, or at least *very* slowly. Is there a reason for this? Could the efficiency be improved somehow?
- In the causal graph, why is race (X) affected by pre-admission statistics (Z)? That seems at least very curious.
- The authors write that "Algorithmic fairness [15, 28, 2, 30] operationalizes evaluation of differential performance of machine learning models across subpopulations" - this is a very reductive description of the AF field. Not all AF conceptions are about subpopulations, and even of those not all are about differential performance. Some would say that AF also encompasses ethical questions beyond those that can be answered by looking at csv files with model predictions and "ground truths". (For one: what the authors are doing in this manuscript does not fall under the above description but would surely be classified as "algorithmic fairness".)

**Ethical Concerns:**

["NO or VERY MINOR ethics concerns only"]

**Final Justification:**

This is a technically solid and very well-written manuscript that includes a non-trivial novel problem framing, theoretical developments in terms of path-specific effect estimators, and a complex, timely, and relevant case study. The original manuscript was already strong, and additionally, the authors have thoroughly engaged with all the points raised by the reviewers, including conducting additional experiments. Hence, I am (still) recommending acceptance.

**Limitations:**

I would like to see the limitations of this approach presented more clearly, specifically the dependence on a rather large range of complex assumptions about the data-generating process that may be difficult to understand in detail and even more difficult to verify for a given application. One obvious risk is that people might misapply this method and conclude that something is "fair", when really it is not.

**Paper Formatting Concerns:**

No formatting concerns.

**Quality:**

4

**Strengths And Weaknesses:**

The manuscript is very well-written, thorough, and dense. The development of a novel, doubly robust path-specific effect estimator in the presence of multiple mediators is an essential contribution and presented very thoroughly. The provided theoretical analysis and empirical experiments (confirming the validity of the theoretical results) further strengthen this contribution. Finally, the case study alone is already very interesting and well-conducted, and moves far beyond the prototypical standard bias case studies presented in many papers. The topic of the case study is also very timely; I believe the manuscript really adds something to the ongoing debate about these measurement discrepancies.

I essentially don't see any major weaknesses - I just have some questions or very minor suggestions which I am leaving below.

---

> ### Author Rebuttal · Authors · 2025-07-31
>
> Thank you for your thoughtful review and for raising constructive points, which we address below.
>
> **Questions**
> 1. In our analysis, $V$ includes $\text{SpO}_2$, $\text{SaO}_2$, and their discrepancy, which is inherently defined by the joint behavior of $\text{SpO}_2$ and $\text{SaO}_2$. Isolating a path solely through the discrepancy is not as meaningful in our setting because race influences $\text{SpO}_2$ and $\text{SaO}_2$ directly, and the discrepancy arises from their measurements, not from how it is mathematically computed. While the VDE may, as the reviewer noted, include components like $\text{SaO}_2$ that are not inherently unfair, we include the full set of oximetry-related variables in $V$ to more comprehensively capture the mechanism by which racial bias in pulse oximetry mediates clinical decisions. We will clarify this modeling choice explicitly in the revised manuscript.
>
> ###
>
> 2. We have previously identified an issue in the hyperparameter search that incorrectly selected the number of estimators for the XGBoost model. After conducting a proper hyperparameter search across 64 settings for each propensity and regression model, both the eICU and MIMIC datasets show a **positive** VDE. Specifically, eICU has an effect of $0.8$ hrs ($95 \\%$ CI [$0.7$ to $0.9$]), while MIMIC shows an effect of $7.4$ hrs ($95 \\%$ CI [$7.2$ to $7.7$]). The results related to invasive ventilation rates are not meaningfully affected. We will update Figure 5b and the corresponding text in the revised manuscript to reflect the corrected estimates and clarify what the VDE measures.
>
> ###
>
> 3. Although the variance still decreases, the convergence rate may be slow because of the propensity score clipping. Nonetheless, we emphasize that the purpose of the self-normalized estimator is to mitigate variance compared to the canonical doubly robust version. Empirically, the behavior is better than one without self-normalization (see Figure 3a vs. 3b). Improving these properties further is certainly an active area of open research more broadly.
>
> ###
>
> 4. We agree that the variable $X$, representing race, should not be modeled as receiving a directed arrow from $Z$ alone. The more appropriate structure includes a hidden confounder between $X$ and $Z$, as demographic variables such as age may influence the distribution of racial groups. Importantly, introducing a bidirected arrow between $X$ and $Z$ does not affect our estimators or the theoretical analysis presented in Appendix A. We will revise the causal graph accordingly in the updated manuscript.
>
> ###
>
> 5. We agree and will expand the definition to the following: "Algorithmic fairness operationalizes computational methodologies to identify, measure, and address disparate behavior related to (automated or other) interventions such as decision-making processes. One prominent line of work evaluates differential performance of machine learning models across subpopulations…"
>
> **Limitations**
> 1. We agree on the importance of clearly stating the assumptions underlying our approach. As with most causal inference methods, our framework relies on a specified causal graph and thus depends on assumptions about the data-generating process that can be challenging to verify in practice. Consequently, it shares common limitations related to potential misspecification of the causal structure. For example, if there is strong reason to believe that $W$ and $V$ have a latent common cause in a given application, the path-specific effect is unidentifiable. While our setup is more general than has been considered in typical causal fairness frameworks and more appropriate for studying bias in pulse oximetry applications, we will clarify these assumptions more explicitly in the "Limitations" section of the manuscript. This includes both assumptions specific to our ICU application in Section 4, and more generally, the limitations of potential conclusions drawn from our framework.
>
> We hope this addresses the points you raised in your review. We are happy to clarify any additional questions you may have.

---

> > ### Comment · Reviewer_1gtf · 2025-08-07
> >
> > I would like to thank the authors for their quite comprehensive responses!
> >
> > My concerns have been well-addressed, and I am keeping my original score (accept).
> >
> > Just one final remark on the causal graph: introducing a confounder "age" between race (X) and pre-admission statistics (Z) would also not seem correct to me - age surely has no causal effect on race. I believe what would be needed would be a selection node that receives directed edges e.g. from race, age, etc.

---

### Decision · Program_Chairs · 2025-09-17

**Decision:**

Accept (poster)

**Comment:**

The paper proposes mediating multiple confounders with path-specific effects to measure bias due to specific confounders. Reviewers generally found the paper interesting and a good contribution over existing work. Some concerns were raised over the assumptions made by the method, and the lack of incorporation of time-series data (e.g. changing code status) is a significant limitation. Otherwise it an interesting approach to modelling racial bias.